# A generally conserved response to hypoxia in iPSC-derived cardiomyocytes from humans and chimpanzees

Michelle C Ward[1]*, Yoav Gilad[1,2]*

[1]Department of Medicine, University of Chicago, Chicago, United States;
[2]Department of Human Genetics, University of Chicago, Chicago, United States

**Abstract** Despite anatomical similarities, there are differences in susceptibility to cardiovascular disease (CVD) between primates; humans are prone to myocardial ischemia, while chimpanzees are prone to myocardial fibrosis. Induced pluripotent stem cell-derived cardiomyocytes (iPSC-CMs) allow for direct inter-species comparisons of the gene regulatory response to CVD-relevant perturbations such as oxygen deprivation, a consequence of ischemia. To gain insight into the evolution of disease susceptibility, we characterized gene expression levels in iPSC-CMs in humans and chimpanzees, before and after hypoxia and re-oxygenation. The transcriptional response to hypoxia is generally conserved across species, yet we were able to identify hundreds of species-specific regulatory responses including in genes previously associated with CVD. The 1,920 genes that respond to hypoxia in both species are enriched for loss-of-function intolerant genes; but are depleted for expression quantitative trait loci and cardiovascular-related genes. Our results indicate that response to hypoxic stress is highly conserved in humans and chimpanzees.
DOI: https://doi.org/10.7554/eLife.42374.001

*For correspondence:
mcward@uchicago.edu (MCW);
gilad@uchicago.edu (YG)

Competing interests: The authors declare that no competing interests exist.

## Introduction

Understanding human susceptibility to disease, and the mechanisms that underlie disease susceptibility, are central goals of biomedical research. One common approach to investigate the regulatory mechanisms that underlie inter-individual disease susceptibility differences is to combine disease association studies with investigations of genetic variants that associate with molecular-level phenotypes using a quantitative trait locus (QTL) framework. However, one of the limitations of a QTL-based approach is that it is not clear what proportion of loci have actual functional consequences. A complementary approach to gaining insight into human susceptibility to disease is to investigate the genetic, molecular, and cellular differences between humans and our closest evolutionary relatives, the great apes. The challenge of a comparative approach is that it can be difficult to determine the specific basis of inter-species phenotypic differences, such as disease, because many observations are often correlated with each other across the species. The limitations and challenges of these approaches may be addressed by a combined analysis of comparative and QTL data, which can help us better understand the functional role of regulatory QTLs, by focusing on QTLs that impact genes whose regulation is either conserved or correlated with inter-species phenotypic differences.

The framework for our comparative study begins with the observed inter-species difference in cardiovascular disease (CVD). CVD is responsible for about a third of both human and captive chimpanzee deaths (WHO; *Varki et al., 2009*). The anatomy of the healthy human and chimpanzee heart is similar; however CVD pathology differs. Chimpanzee disease is often associated with interstitial myocardial fibrosis, while human heart disease predominantly results from coronary artery arthero-sclerosis, leading to ischemic damage (*Lammey et al., 2008*; *Varki et al., 2009*). The interstitial fibrosis etiology prevalent in chimpanzees is also the most frequently diagnosed form of CVD in

**eLife digest** Understanding why some people get heart disease and others do not could help scientists find better ways to treat or prevent the condition. Genetics likely plays a role, and one way to identify genes that are important for heart health is to compare genes in humans and their closest evolutionary relatives, the chimpanzees. Though it is not exactly the same as seen in humans, chimpanzees do get heart disease. Differences in the genes involved in heart disease in humans and chimpanzees may help explain what leads to the disease in humans.

Studying heart disease in chimpanzees and humans has been challenging because heart tissue from humans and chimpanzees is hard to come by. Yet scientists can now convert easy-to-access skin cells from humans and chimpanzees into heart cells and grow them under laboratory conditions.

Ward and Gilad have used exactly this approach to see how human and chimpanzee cells respond when they are starved of oxygen, which simulates a heart attack. First, skin cells collected from eight humans and seven chimpanzees were coaxed into becoming heart cells and grown in the laboratory. Ward and Gilad then compared the activity levels of about 12,000 genes in these heart cells when their oxygen was limited. The responses were very similar, with 1,920 genes switching on or off in both species. But the activity of hundreds of other genes differed between humans and chimpanzees. For example, a gene called *RASD1*, which is known to be important in human heart disease, became active in oxygen-starved human cells but not in chimpanzee cells.

Genes that vary in their activity between healthy human individuals are thought to be important in disease. However, Ward and Gilad found that the activity of genes that switch on or off in both species after oxygen starvation did not vary a lot in a collection of heart samples from hundreds of individuals. These experiments may help scientists narrow down which genes are likely most important in heart disease. More studies are needed to understand what these genes do and how they contribute to heart disease.

DOI: https://doi.org/10.7554/eLife.42374.002

captive bonobos, gorillas and orangutans (*Lowenstine et al., 2016*). It is unclear whether these apparent differences in susceptibility between species are due to genetic or environmental factors.

A consequence of myocardial ischemia, the reduction of blood flow to the heart tissue, is oxygen deprivation. Oxygen sensing and response is an essential process across species. If the balance between anti-oxidants and pro-oxidants, such as reactive oxygen species (ROS), is decoupled, redox signaling is disrupted, and oxidative stress ensues. The heart is the most oxygen-demanding tissue in the body (*Giordano, 2005*). Maintenance of oxygen homeostasis is essential for cardiac function as imbalance of ROS can result in myocardial infarction and heart failure. Indeed, 20–40 min of oxygen deprivation results in irreparable damage to the human heart (*Bretschneider et al., 1975*).

It is well appreciated that CVD is a complex disease with many contributing genetic and environmental factors. These effects are difficult to distinguish in in vivo studies within and between species because, in order to establish clear causal relationships and mechanism, directed perturbation is required. This is infeasible in humans and other apes due to practical and ethical considerations. More tractable model organisms such as mice are not optimal models of CVD given the differences in relative heart size and heart rate (*Doevendans et al., 1998*; *Milani-Nejad and Janssen, 2014*). In addition to genome-level differences between humans and mice, the electrophysiology of mouse cardiomyocytes differs substantially from that of human cardiomyocytes (*Moretti et al., 2013*). To understand intrinsic gene regulatory processes in cell types relevant to human disease, one might have to study human cells. Primary human cardiac cells are not easy to access, and have a limited lifespan in cell culture. The advent of induced pluripotent stem cell (iPSC) technology now allows us to access disease-relevant cell types across human individuals and other primates, control the extra-cellular environment of these cells in culture, and test the effects of perturbation. We have recently established a panel of human and chimpanzee iPSC lines (*Gallego Romero et al., 2015*), and we have shown that cardiomyocytes derived from induced pluripotent stem cells (iPSC-CMs) can effectively model gene regulation in hearts from humans and chimpanzees (*Pavlovic et al., 2018*). iPSC-CMs can be used to study CVD phenotypes including channelopathies such as long QT syndrome, and dilated cardiomyopathies such as Barth syndrome (*Tanaka et al., 2015*).

Cardiomyocytes make up 70–85% of the heart volume, 30–40% of the total cellular composition (*Pinto et al., 2016*; *Zhou and Pu, 2016*), and are susceptible to ischemia following coronary artery occlusion. In order to gain insight into human gene regulatory adaptation in the heart, and the evolution of disease susceptibility and resistance, we developed a model of hypoxia and re-oxygenation in human and chimpanzee iPSC-CMs. This cell culture-based system enables an in-depth characterization of the inter-species response to, and recovery from, hypoxic stress. We can now determine intrinsic inter-species regulatory differences in response to a universal cellular stress, which could provide insight into the observed phenotypic differences in the manifestation of CVD between species.

Hypoxia induces a transcriptional response following stabilization of the HIF transcription factors under conditions of low oxygen (*Samanta and Semenza, 2017*). We therefore determined both the global transcriptional response to hypoxia and re-oxygenation by RNA-seq, and the cellular response by measuring features of oxidative damage including lipid peroxidation and DNA damage, cytotoxicity, and cytokine release, in both species. While an iPSC-derived cardiomyocyte-based system has been previously used to study the effects of hypoxia in a single human and a single rhesus macaque individual, here we use a panel of human and chimpanzee individuals to identify a set of conserved and species-specific response genes (*Zhao et al., 2018*). The identification of inter-species gene regulatory differences allowed us to develop hypotheses about molecular mechanisms that might explain phenotypic differences between species.

## Results

We differentiated cardiomyocytes from iPSCs of eight human and seven chimpanzee individuals, including replicate differentiations from a subset of the lines (*Figure 1A*, and *Figure 1—figure supplement 1A*). The 15 human and chimpanzee iPSC lines we used have been well characterized as described here and previously (*Figure 1—figure supplement 2* and Key Resource Table) (*Gallego Romero et al., 2015*; *Burrows et al., 2016*; *Pavlovic et al., 2018*; *Ward et al., 2018*). To increase the purity of the cardiomyocyte cultures, we used a metabolic purification step (*Tohyama et al., 2013*). To obtain more mature cardiomyocytes, we cultured the cells for 30 days post induction of differentiation (*Chan et al., 2013*; *Robertson et al., 2013*), subjected the cells to electrical stimulation to increase cellular elongation and improve calcium handling (*Chan et al., 2013*), and cultured the cells in the presence of galactose instead of glucose to shift the cells' metabolism from fetal-associated glycolysis to adult-associated mitochondrial respiration (*Rana et al., 2012*).

After 30 days in culture, differentiated cells, likely predominantly ventricular in subtype (*Burridge et al., 2014*), express the cardiomyocyte markers cardiac troponin T (TNNT2), sarcomeric alpha-actinin (ACTN1), and the ventricle-specific marker Iroquois Homeobox 4 (IRX4), in both species (*Figure 1—figure supplement 1B*). We determined the purity of differentiated cardiomyocytes in each culture independently, by measuring the proportion of cells expressing TNNT2 by flow cytometry (see Materials and methods). Only samples with greater than 45% TNNT2-expressing cells (see Materials and methods) were retained. Importantly, there is no difference in the median purity of the cardiomyocyte cultures between humans and chimpanzees (median purity in both species: 78%, *Figure 1—figure supplement 3*). Though we only measured purity by flow cytometry analysis once, we note that *TNNT2* mRNA expression levels, a proxy for purity, do not change during the course of the experiment.

In order to mimic in vivo physiological oxygen levels experienced by cardiomyocytes in the heart, we cultured our cells at 10% oxygen, starting on the 25th day of differentiation. Peri-cellular oxygen levels were measured non-invasively using an oxygen sensor applied to the inside sidewall of the cardiomyocyte cultures. Transferring the cardiomyocytes from atmospheric oxygen levels (21% $O_2$) to 10% $O_2$ did not seem to induce stress in the cultures (*Figure 1—figure supplement 4A*). In what follows, we consider 10% $O_2$ to be the baseline normoxia condition (designated as condition A). In order to determine the response of the cultures to hypoxia and subsequent recovery to normoxia, we subjected the cardiomyocyte cultures to the following conditions (see *Figure 1A* for a schematic of the study design): We first lowered the $O_2$ levels to 1% for 6 hr (condition B), we then re-oxygenated to 10% $O_2$ for 6 hr of recovery in normoxic conditions (condition C), and 24 hr of recovery

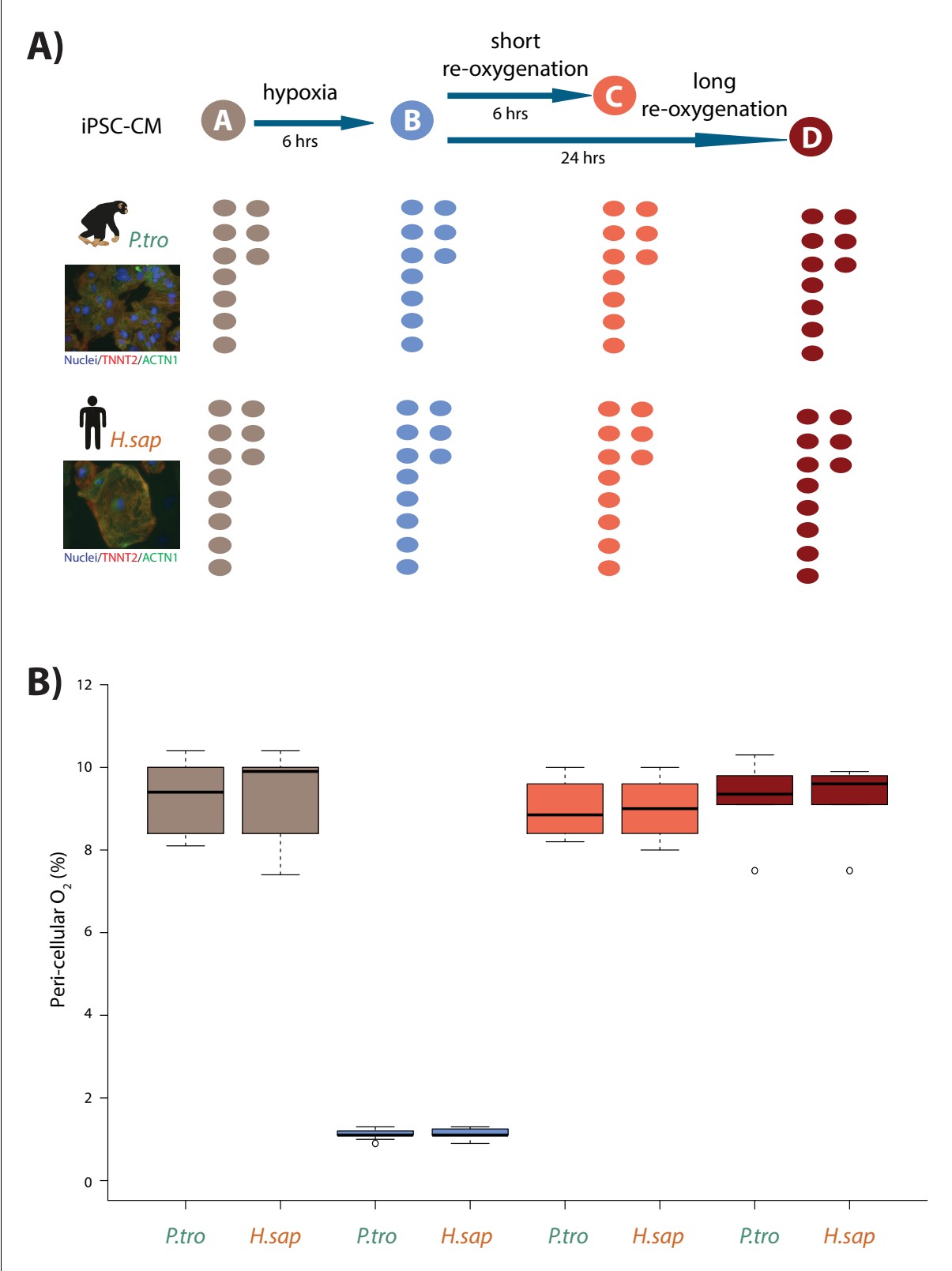

**Figure 1.** Induction of hypoxia in human and chimpanzee iPSC-CMs. (**A**) Experimental design of the study. Cardiomyocytes were differentiated from iPSCs from eight human (*H.sap*), and seven chimpanzee (*P.tro*) individuals together with replicates from three individuals of each species. For the oxygen stress experiment iPSC-CMs in each species were cultured in normoxic conditions (10% $O_2$ - condition A) for 6 hr prior to induction of hypoxia

*Figure 1 continued on next page*

*Figure 1 continued*

at 1% $O_2$ for 6 hr (condition B). Following hypoxia, iPSC-CMs were re-oxygenated to 10% $O_2$ for 6 hr (condition C), or 24 hr (condition D). (B) Peri-cellular $O_2$ levels measured at each stage of the experiment for each experimental batch. Also see *Figure 1—figure supplements 1–5*.

DOI: https://doi.org/10.7554/eLife.42374.003

The following figure supplements are available for figure 1:

**Figure supplement 1.** Cardiomyocytes can be differentiated from iPSCs from both humans and chimpanzees.

DOI: https://doi.org/10.7554/eLife.42374.004

**Figure supplement 2.** New human iPSC lines are pluripotent and display normal karyotypes.

DOI: https://doi.org/10.7554/eLife.42374.005

**Figure supplement 3.** iPSC-CM purity is similar in humans and chimpanzees.

DOI: https://doi.org/10.7554/eLife.42374.006

**Figure supplement 4.** Optimising the induction of a hypoxic response in iPSC-CMs.

DOI: https://doi.org/10.7554/eLife.42374.007

**Figure supplement 5.** Hypoxia induces oxidative damage in both species.

DOI: https://doi.org/10.7554/eLife.42374.008

(condition D). Oxygen levels were monitored and recorded for each experimental batch (*Figure 1B*). The same differentiation culture was used across conditions for each individual.

To confirm that perturbing the peri-cellular oxygen level affected cardiomyocytes from both species, we measured two phenotypes associated with oxidative damage. First, we determined whether damage to DNA is induced following hypoxic stress. Guanine is the nucleotide most prone to oxidation, and in general undergoes base excision repair. The repair products of the oxidative DNA lesions are subsequently released as 8-hydroxydeoxyguanosine (8-OHdG). The amount of 8-OHdG released therefore reflects both the amount of oxidative damage to DNA, and the efficiency of base excision repair. We observed an increase in the level of 8-OHdG following hypoxia and re-oxygenation in both species ($B_{Mean}$ = 0.09 vs. $C_{Mean}$ = 1.32; t-test; p=0.002 in chimpanzees, and $B_{Mean}$ = −0.21 vs. $C_{Mean}$ = 1.42; p=0.01 in humans; *Figure 1—figure supplement 5A*). Within a condition, there is no difference in the amount of 8-OHdG released between species (human $D_{Mean}$ = 1.19, chimpanzee $D_{Mean}$ = 1.34). Second, we determined the extent of lipid peroxidation by measuring the release of the isoprostane 8-iso-Prostaglandin F2α (8-iso-PGF2α), which is induced following ROS-mediated damage to cellular membrane phospholipids. We found an increase in 8-iso-PGF2α release in chimpanzees following hypoxia ($A_{Mean}$ = 0 vs. $B_{Mean}$ = 35.65; p=0.006), and a further increase following long-term re-oxygenation ($D_{Mean}$ = 77.28; p=0.03; *Figure 1—figure supplement 5B*). While there is no difference in 8-iso-PGF2α release between species within any condition, we do not find a significant increase in 8-iso-PGF2α release following hypoxia in humans ($A_{Mean}$ = 0 vs. $B_{Mean}$ = 13.25; p=0.2). This pattern may be explained by incomplete power to detect differences in 8-iso-PGF2α release either between time points or between species. Nevertheless, our observations are intriguing because 8-iso-PGF2α is known to be elevated in heart failure (*Mallat et al., 1998*; *Wolfram et al., 2005*), and is a risk marker for coronary heart disease (*Schwedhelm et al., 2004*).

## Characterizing the regulatory response to hypoxia in cardiomyocytes

We used RNA sequencing to characterize gene expression levels in all conditions, and study the regulatory response to hypoxia in the human and chimpanzee cardiomyocytes (see Materials and methods). We processed the samples using a study design that was balanced with respect to a number of recorded potential technical confounders (*Supplementary file 1*-Table S1). Following sequencing of the RNA, we mapped reads to primate orthologous exons (*Figure 2—figure supplement 1*), and filtered lowly-expressed genes to yield a final data set with expression measurements for 11,974 genes (see Materials and methods). Within each condition, inter-species correlation in read counts is somewhat lower than intra-species variation, as expected (median Spearman's correlation when comparing human samples = 0.97, when comparing chimpanzee samples = 0.98, and for human vs. chimpanzee samples = 0.92; *Figure 2—figure supplement 2*). Using the RNA-seq data we confirmed that genes known to be expressed in cardiomyocytes are expressed in both our human and chimpanzee samples, including genes involved in cardiac structure, ion channels, and adrenoreceptors (*Figure 2—figure supplement 3*).

As mentioned above, we included in our study differentiation replicates from a subset of samples (see *Supplementary file 1*-Table S2 for details). We expect that gene expression data from pairs of replicates should be more similar to each other than to data from any other individual. We used this expected property of the data to account and correct the entire data set for unwanted technical variation (see Materials and methods for more details). After accounting for unwanted variation, samples cluster by species and then by individual or condition (*Figure 2—figure supplement 4*). We note that one technical factor, the presence or absence of episomal reprogramming vectors (three human samples tested positive; *Figure 2—figure supplement 5*), remains a partial confounder with species. However, we confirmed that our conclusions are robust with respect to the inclusion of these three human samples (*Figure 2—figure supplement 6* and *Supplementary file 1*-Table S3).

To determine which genes respond to hypoxia, we analyzed the data from all four conditions using the framework of a linear model. The model included fixed effects for 'species', 'condition', a 'species by condition interaction', a random effect for 'individual', and four unwanted factors of variation as covariates (see Materials and methods). For this analysis, we randomly selected one of each of the samples we had replicate data for. We were first interested in classifying genes into the following four categories within each species independently: genes that respond to hypoxia, genes that respond to short-term (6 hr) re-oxygenation following hypoxia, genes that respond to long-term (24 hr) re-oxygenation following hypoxia, and genes that differ between long-term re-oxygenation and baseline normoxia. Of 11,974 expressed genes, we identified ~4,000 genes that respond to hypoxia at 10% FDR in each species, and a slightly higher number of genes whose expression has changed upon re-oxygenation (*Figure 2*; the results of all tests are in *Supplementary file 1*-Table S3A).

We then focused on inter-species gene expression differences within each condition, independently, and found that roughly half of all expressed genes are differentially regulated between species, regardless of the condition (at FDR of 10%; *Supplementary file 1*-Table S3A, *Figure 2—figure supplement 7A*). Using this approach we were also able to identify hundreds of genes that are differentially expressed between species exclusively in a single condition, for example following hypoxia (*Figure 2—figure supplement 7B*). However, this approach does not provide strong evidence for true differences in the dynamic response to hypoxia between humans and chimpanzees, because of incomplete power to detect inter-species differentially expressed genes in any given condition. Thus, in order to determine the species-specificity of the global response to changing oxygen conditions we explicitly compared the effect size of expression change between pairs of conditions, for all genes, across species. Overall, there is a strong correlation in the global gene expression response to both hypoxia and re-oxygenation in humans and chimpanzees (median Spearman correlation = 0.78; sign test $p<10^{-4}$ for all comparisons; *Figure 3*), suggesting that the general response to changes in oxygen level is conserved in the two species. Genes that respond to either hypoxia or re-oxygenation in both species include *VEGFA* (a known hypoxia response gene), *TRPV1* (implicated in ischemia-reperfusion injury in the heart [*Wang and Wang, 2005*]), and *DDX41* (implicated in stress survival regulation [*Shih and Lee, 2014*]).

## Species-specific transcriptional changes in response to hypoxia

The observation that the response to changes in oxygen level is generally conserved in the two species notwithstanding, we next focused on dynamic inter-species differences in our study. To do so, we used two approaches. First, we estimated a gene-specific interaction effect between species using the framework of the linear model described above. We identified 147 genes that responded to hypoxia in one species but showed little or no response in the other species, or that responded in both species but showed the opposite direction of effect (at FDR of 10%; *Figure 4*; *Supplementary file 2*). We did not find inter-species differences in the response to either the short or long re-oxygenation treatments.

We did not find enrichment of particular pathways among the species-specific response genes, but this may not be surprising as stress response pathways are often regulated by a small number of key master regulator genes (*Haynes et al., 2010*; *Li et al., 2011*; *Natarajan et al., 2013*; *Mahat et al., 2016*; *Quirós et al., 2017*). However, several of the genes with significant species by condition interactions have functions related to G-protein signaling, TGF-β signaling, and metabolism (*Figure 4—figure supplement 1*). Our most significant interaction corresponds to the *RASD1* gene, which is up-regulated specifically in humans following hypoxia (*Figure 4*). This gene encodes a

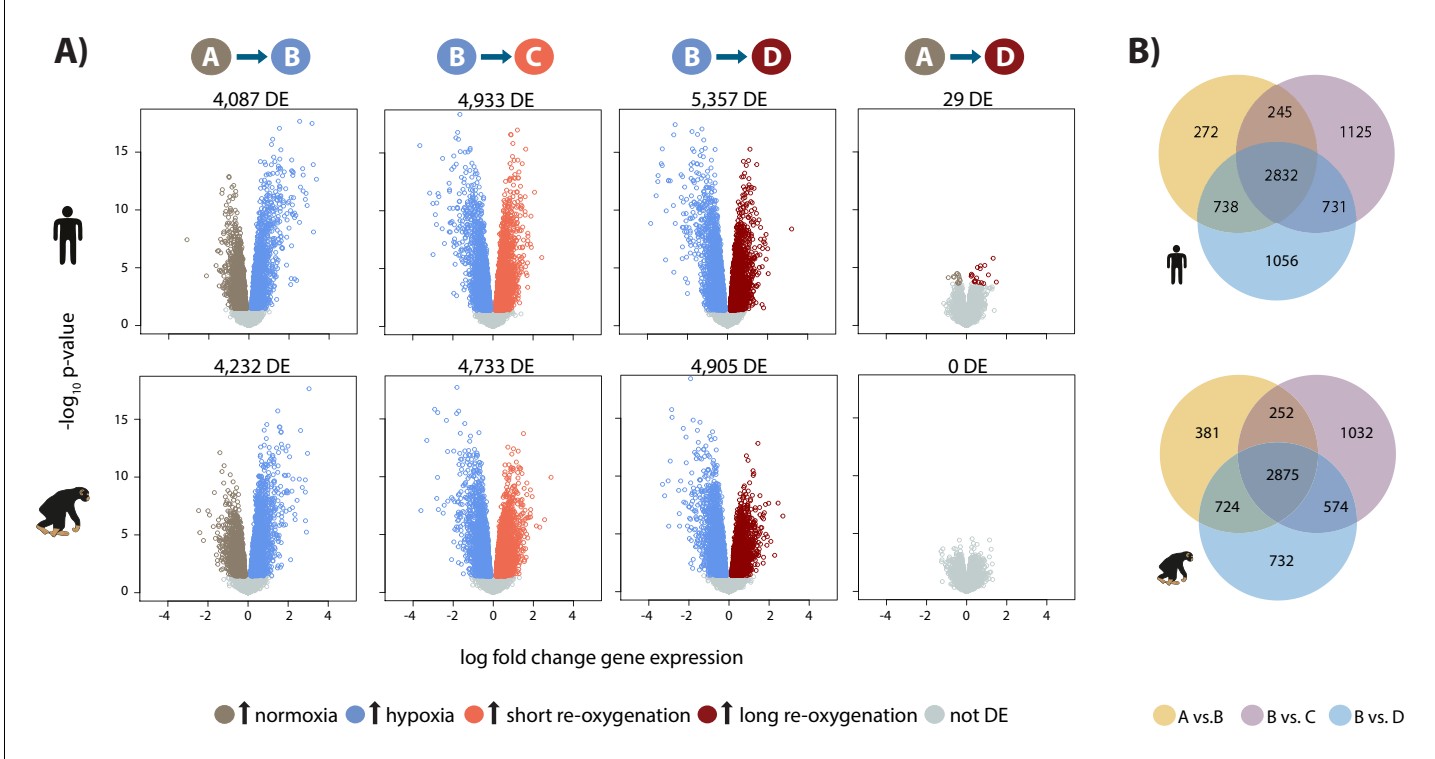

**Figure 2.** Hypoxia induces a gene expression response in humans and chimpanzees. (**A**) Volcano plots representing genes that are differentially expressed (DE; 10% FDR) in pairwise comparisons across conditions in each species independently. In a comparison of A vs. B, genes that are up-regulated in hypoxia are represented in blue, and genes that are up-regulated in normoxia are represented in brown. Genes that are up-regulated in condition C are represented in coral, and genes that are up-regulated in D are represented in dark red. (**B**) Overlap of genes that are differentially expressed in pairs of conditions in each species independently. Also see **Figure 2—figure supplements 1–7**.

DOI: https://doi.org/10.7554/eLife.42374.009

The following figure supplements are available for figure 2:

**Figure supplement 1.** RNA-seq sample quality is similar between species.
DOI: https://doi.org/10.7554/eLife.42374.010
**Figure supplement 2.** Inter-species variability in read counts is greater than intra-species variability.
DOI: https://doi.org/10.7554/eLife.42374.011
**Figure supplement 3.** Range of cardiomyocyte genes are expressed in human and chimpanzee iPSC-CMs.
DOI: https://doi.org/10.7554/eLife.42374.012
**Figure supplement 4.** RNA-seq samples cluster by species and then by oxygen level or individual.
DOI: https://doi.org/10.7554/eLife.42374.013
**Figure supplement 5.** Species and individual are most correlated with the first two principal components in PCA.
DOI: https://doi.org/10.7554/eLife.42374.014
**Figure supplement 6.** Inter-species results are recapitulated using a subset of the data.
DOI: https://doi.org/10.7554/eLife.42374.015
**Figure supplement 7.** Thousands of genes are differentially expressed between species in each condition.
DOI: https://doi.org/10.7554/eLife.42374.016

Ras GTPase, which activates G-protein signaling. Conversely, the *LRRC25* gene responds to hypoxia specifically in chimpanzees, and has been found to inhibit NF-κβ signaling (*Feng et al., 2017*) (*Figure 4*).

Directly modeling interaction effects with small numbers of samples is an underpowered approach. In order to side-step the challenge of incomplete power when performing multiple pair-wise comparisons, we used a second approach; a joint Bayesian model, to classify genes based on their expression levels between conditions within each species during the course of the hypoxia-re-oxygenation experiment (see Materials and methods). Four gene clusters were empirically deter-mined to explain the predominant expression patterns in the data (lowest BIC and AIC after testing

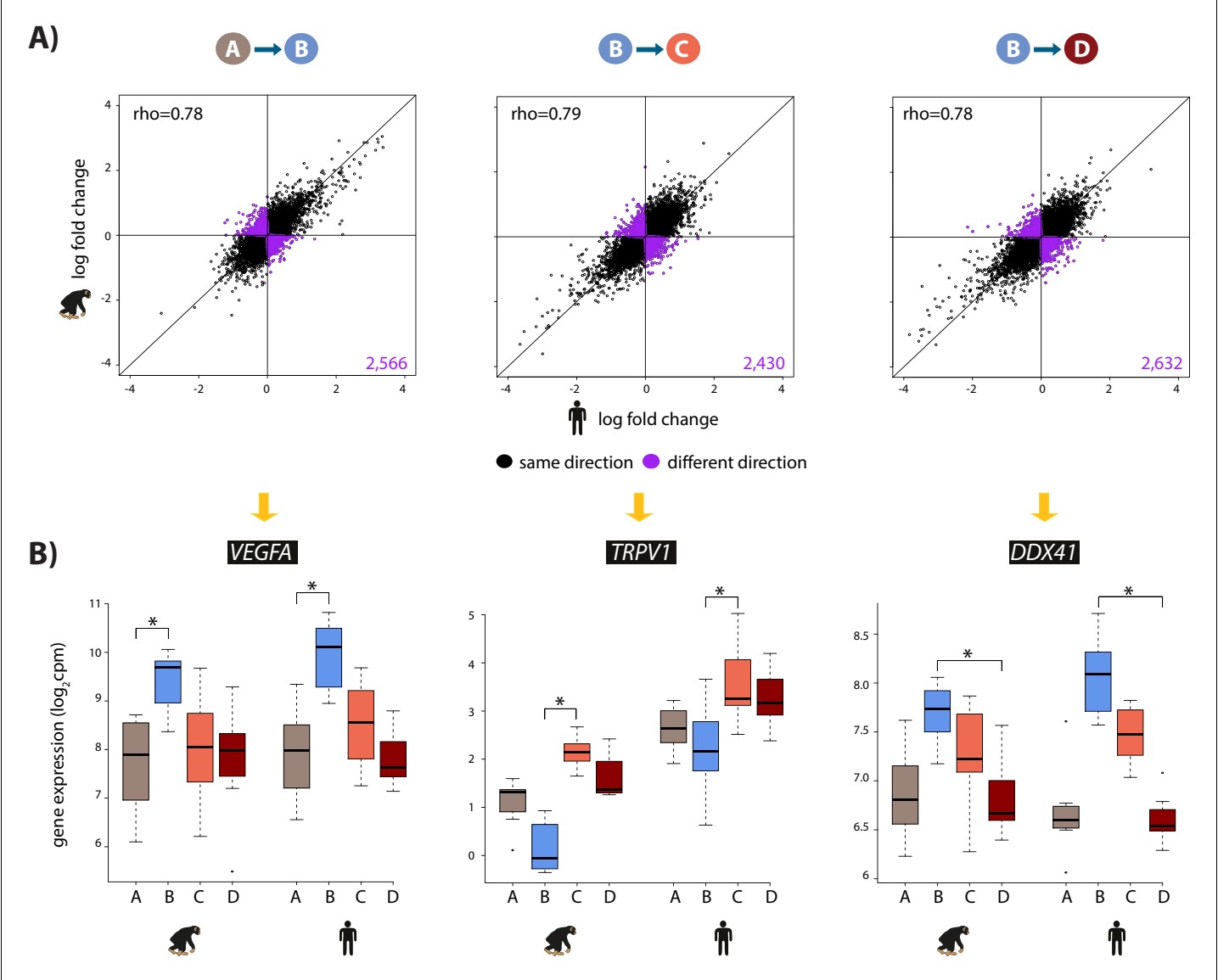

**Figure 3.** The hypoxic gene expression response is highly correlated across species. (**A**) The log fold change in expression of 11,974 genes between pairs of conditions in humans on the x-axis, and chimpanzees on the y-axis. Genes whose expression changes in the same direction in both species are represented in black, and genes whose expression change direction differs across species are represented in purple. (**B**) Examples of genes that are differentially expressed in both species in A vs. B (*VEGFA*), B vs. C (*TRPV1*), and B vs. D (*DDX41*) are shown. Asterisk denotes a statistically significant difference in expression between conditions (10% FDR).

DOI: https://doi.org/10.7554/eLife.42374.017

1–15 clusters; *Figure 5—figure supplement 1*). Using this approach we categorized 9,414 genes as not responding to hypoxia in either humans or chimpanzees (non-response genes), 1,920 genes that respond to hypoxia in both species (conserved response genes), 430 genes that respond to hypoxia in chimpanzees only (chimpanzee-specific response genes), and 199 genes that respond to hypoxia only in humans (human-specific response genes; *Figure 5* and *Supplementary file 3*). It is notable that there is no prevalent pattern of genes responding specifically to re-oxygenation in either species, which suggests that the expression of most genes returns to baseline by the end of the experiment. We do not identify additional gene expression patterns even when we increase the number of clusters.

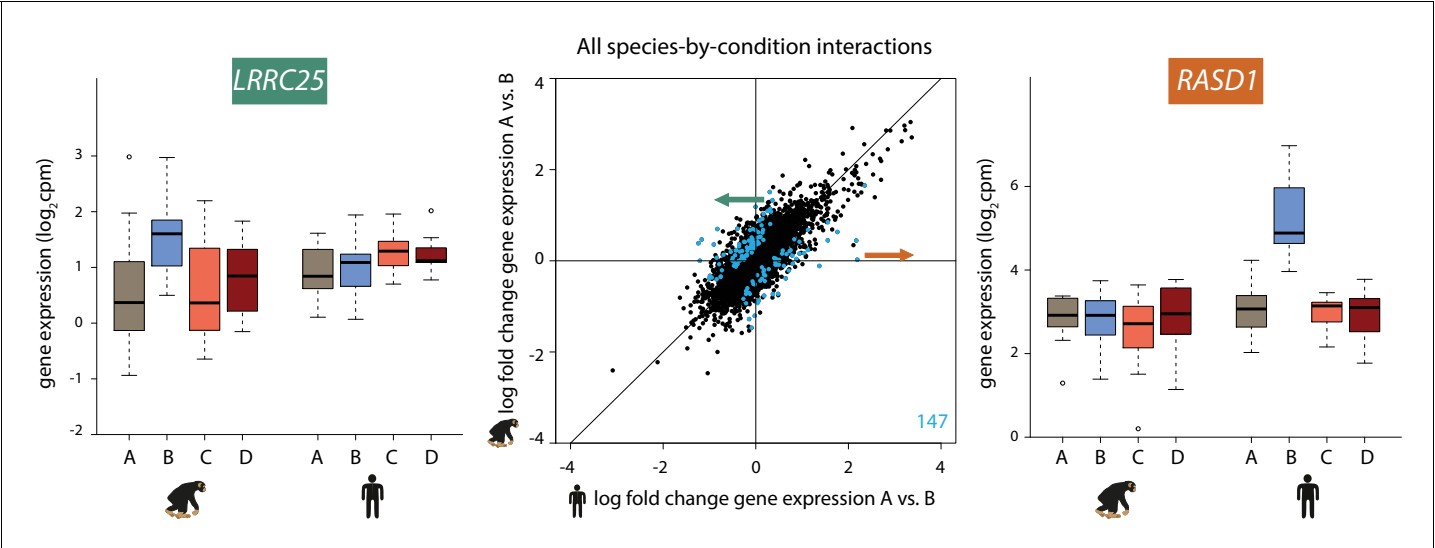

**Figure 4.** 147 genes show a species-specific response following hypoxia. Middle panel: The log fold change in expression of 11,974 genes between normoxia (A) and hypoxia (B) in humans on the x-axis, and chimpanzees on the y-axis. Genes with a species-by-condition interaction are represented in blue. All species-by-condition interactions affected the hypoxic condition; therefore only a representative pairwise comparison is shown (A vs. B). An example of a gene that responds in chimpanzees only (*LRRC25*) is shown in the left panel, and an example of a gene that responds only in humans (*RASD1*) is shown in the right panel. Also see *Figure 4—figure supplement 1*.

DOI: https://doi.org/10.7554/eLife.42374.018

The following figure supplement is available for figure 4:

**Figure supplement 1.** Genes responding in a species-specific manner have a variety of cellular functions.

DOI: https://doi.org/10.7554/eLife.42374.019

## Properties of hypoxic response genes

To confirm that our approach identifies meaningful response genes, we considered the overlap of genes assigned to our four response categories with a set of genes previously identified to respond to hypoxia in human, and a more evolutionary distant primate, the rhesus macaque (*Zhao et al., 2018*). As expected, genes previously found to respond to hypoxia are enriched among genes assigned to the 'conserved response' category in our study, and depleted among genes assigned to the 'non-response' category (Chi-squared test; $p < 10^{-15}$ in both; *Figure 6—figure supplement 1*).

To explore properties of the response genes, we integrated our gene expression data with data from human chromatin immunoprecipitation followed by high through-put sequencing (ChIP-seq) experiments for three transcription factors that are known to bind to the genome in response to altered oxygen levels - HIF1α, HIF2α and FOXO3 (*Schödel et al., 2011*; *Eijkelenboom et al., 2013*). We arbitrarily designated genes as potentially regulated by the three transcription factors by identifying the closest orthologous gene to each human transcription factor-bound region. Consistent with previous literature (*Samanta and Semenza, 2017*), the 356 HIF1α-bound regions, and 301 HIF2α-bound regions are enriched near conserved response genes compared to non-response genes (Chi-squared test; $p < 10^{-10}$ for both factors; *Figure 6A*). We thus asked whether differences in HIF binding could account for inter-species gene expression differences. Indeed, both HIF1α- and HIF2α-bound regions are enriched near conserved response genes compared to species-specific response genes ($p < 0.01$ for both factors; *Figure 6A*). In particular, chimpanzee-specific response genes are depleted for HIF binding ($p < 0.05$ for both factors; *Figure 6A*). The 934 FOXO3-bound regions are not enriched near conserved response genes compared to non-response genes, nor are conserved response genes enriched compared to species-specific response genes.

We asked whether differences in sequence conservation at transcription factor-bound regions are associated with inter-species gene expression differences. To do so, we calculated the phyloP score at each bound region in close proximity to expressed genes. We found no difference in sequence conservation at HIF1α-, HIF2α- and FOXO3-bound regions when comparing conserved response genes to non-response genes, or conserved response genes to species-specific response genes

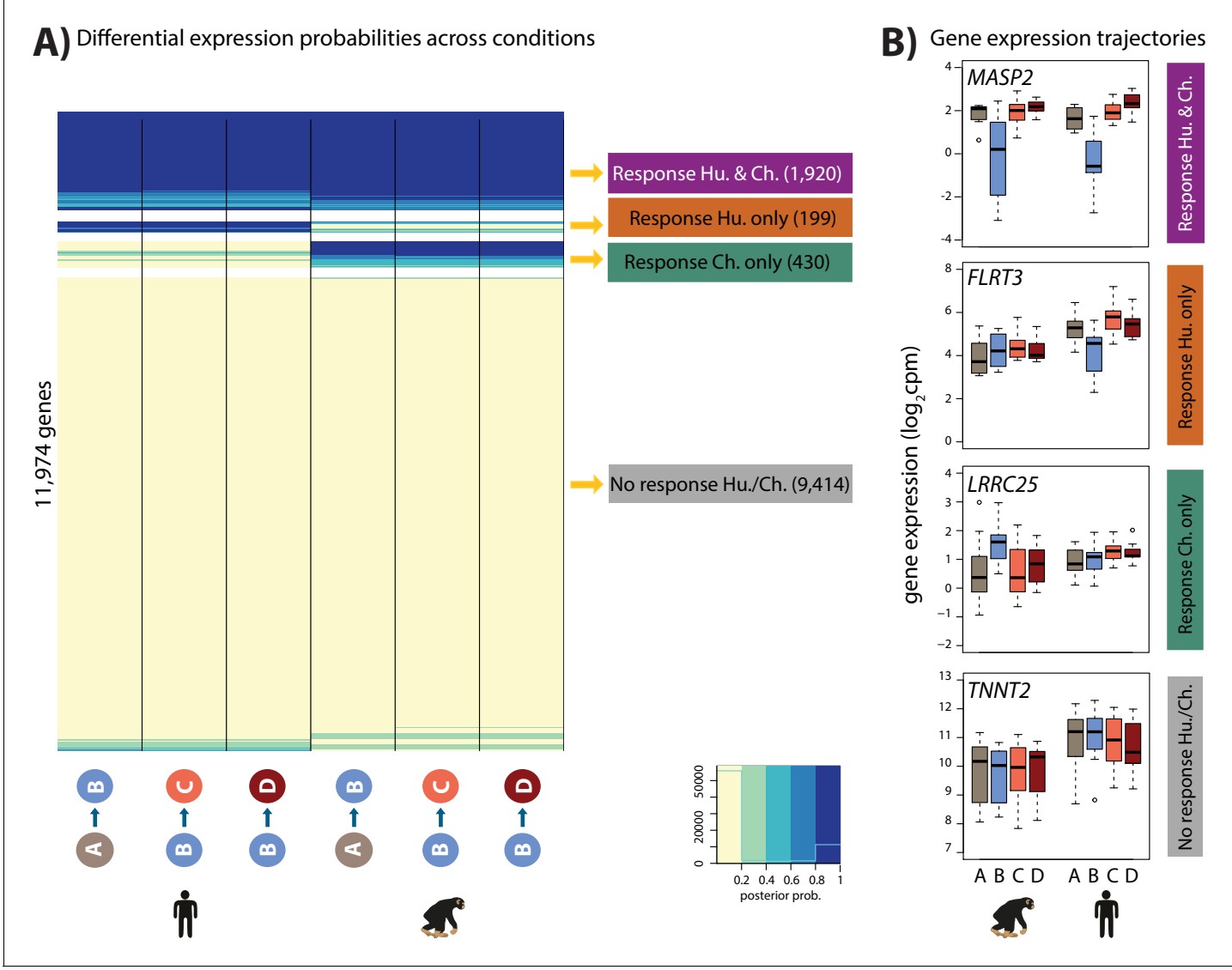

**Figure 5.** Four gene expression response categories can be classified following hypoxia. (**A**) Posterior probabilities of genes being differentially expressed across pairs of conditions. Genes are categorized based on their posterior probabilities: genes with a p<0.5 across all tests are designated as 'Non-response genes' (grey), genes with a p>0.5 across all tests are designated as 'Conserved response genes' (magenta), genes with p>0.5 in human comparisons only are designated as 'Human-specific response genes' (orange), and genes with p>0.5 in chimpanzee comparisons only are designated as 'Chimpanzee-specific response genes' (green). (**B**) Examples of genes belonging to each of the four categories. Also see *Figure 5— figure supplement 1*.

DOI: https://doi.org/10.7554/eLife.42374.020

The following figure supplement is available for figure 5:

**Figure supplement 1.** Four patterns of gene expression predominate across the course of the experiment.
DOI: https://doi.org/10.7554/eLife.42374.021

(*Figure 6B*). However, it should be noted that the numbers of species-specific response genes in proximity to binding sites for these transcription factors is small.

In addition to transcription factor-mediated gene expression responses to stress, non-coding transcripts can also play a regulatory role in hypoxia, immune responses, and cardiac development and disease (*Scheuermann and Boyer, 2013*; *Choudhry et al., 2014*; *Danko et al., 2018*). To characterize the contribution of non-coding transcripts to the hypoxic response, we classified our 11,974 expressed genes as protein-coding, antisense, or long interspersed non-coding RNA (lincRNA; see Materials and methods). The 205 antisense transcripts are enriched in conserved response genes

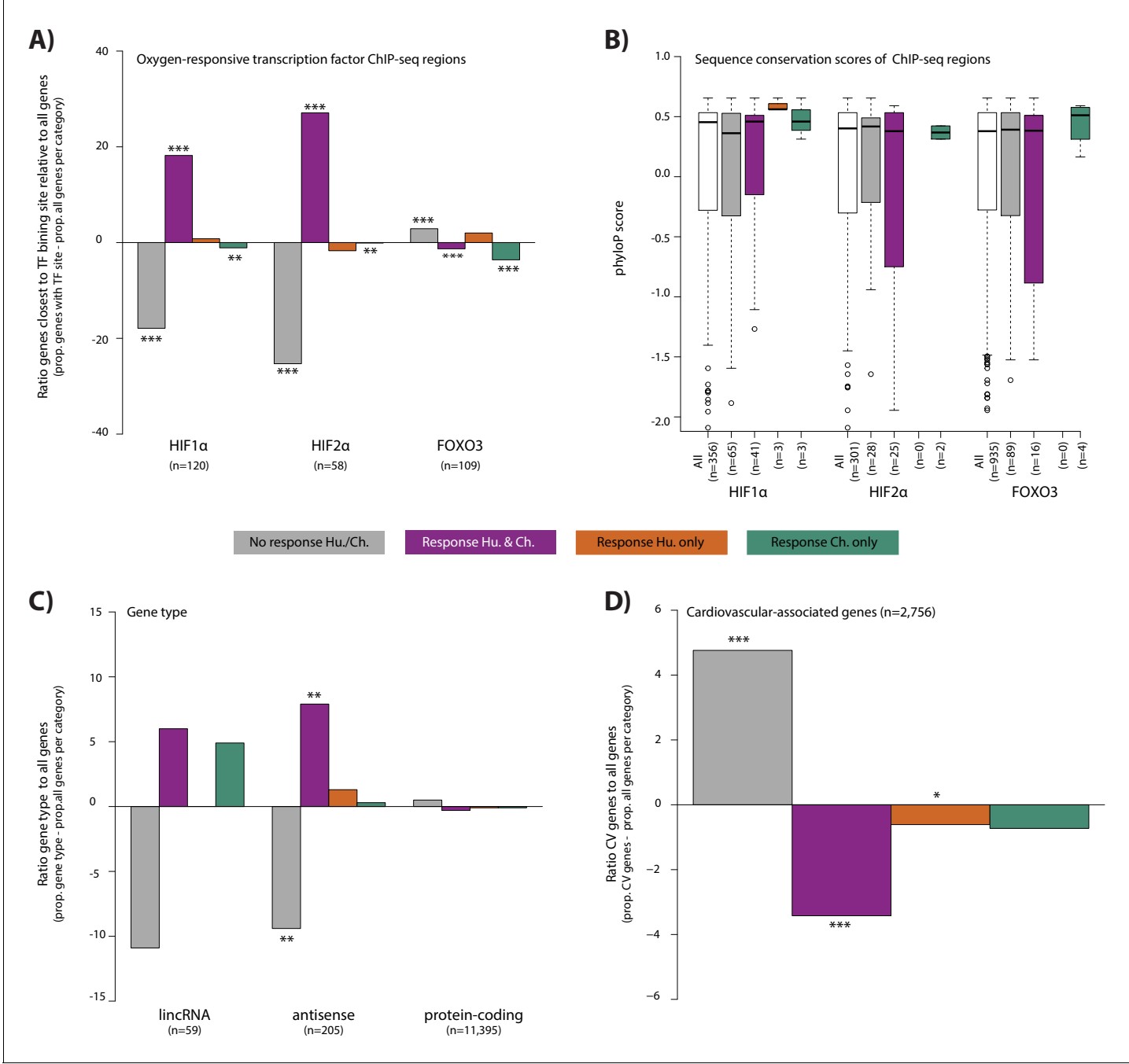

**Figure 6.** Conserved hypoxic response genes are enriched for nearby HIF transcription factor binding locations, and depleted for cardiovascular-associated genes. (A) The proportion of genes close to HIF1α, HIF2α and FOXO3 transcription factor binding locations (Schodel et al., and Eijkelenboom et al.) in each response category relative to the proportion of all genes within a response category. (B) The sequence conservation score (phyloP) of each transcription factor binding location stratified by their proximity to response genes. (C) The proportion of genes annotated as lincRNAs, antisense transcripts and protein-coding genes in each response category relative to the proportion of all genes within a response category. (D) The proportion of cardiovascular-associated (CV) genes (Cardiovascular GO Annotation Initiative) in each response category relative to the proportion of all genes within a response category. Response categories are: non-response (grey), conserved response (magenta), human-specific response (orange), and chimpanzee-specific response (green). Asterisk denotes a significant difference between the proportion of the gene set in each response category, and the proportion of all genes within a response category (Chi-squared test; *p<0.05, **p<0.005, ***p<0.0005). Also see *Figure 6—figure supplements 1–2*.

DOI: https://doi.org/10.7554/eLife.42374.022

The following figure supplements are available for figure 6:

*Figure 6 continued on next page*

*Figure 6 continued*

**Figure supplement 1.** Conserved response genes are enriched in an orthogonal hypoxia data set.
DOI: https://doi.org/10.7554/eLife.42374.023
**Figure supplement 2.** Conserved response genes are enriched in stress response pathways.
DOI: https://doi.org/10.7554/eLife.42374.024

compared to non-response genes (49 vs. 142; p=0.001; *Figure 6C*). However, species-specific response genes are no more likely to be antisense transcripts than conserved response genes. Conserved response genes are no more likely to be one of 59 annotated lincRNAs than non-response genes, and species-specific response genes are no more likely to be lincRNAs than conserved response genes. That said, considering the more stringent set of species-by-condition interaction genes, we found enrichment of lincRNAs compared to all expressed genes (4/147 interaction genes; Fisher's exact test, p=0.007). These lincRNAs are *APTR*, *NEAT1*, *RNF139-AS1* and *LINC02615*.

Finally, we wanted to determine whether response categories are enriched for particular pathways, using a background set of all expressed genes. In the non-response gene category, there is a significant enrichment in KEGG pathways related to the heart (e.g. dilated cardiomyopathy, hypertrophic cardiomyopathy, arrhythymogenic right ventricular cardiomyopathy and adrenergic signaling in cardiomyocytes, 10% FDR; *Figure 6—figure supplement 2*). In the conserved response category, various signaling pathways related to sensing the external environment, and responding to oxygen are significantly enriched including HIF1$\alpha$, MAPK and FOXO1. There are no significantly enriched pathways in the species-specific gene response categories. Given the apparent enrichment of cardiovascular genes in the non-response category, we explicitly tested the contribution of a set of cardiovascular-associated genes to the response to hypoxia (see Materials and methods). Indeed, we found that there is a depletion of genes implicated in cardiovascular development and disease amongst the genes that respond to hypoxia in both species (Chi-squared test; p=8.3×10$^{-6}$; *Figure 6D*).

## Associating genetic variation with hypoxic response genes

It has been suggested that genetic variants that associate with gene expression levels (eQTLs), may mediate disease phenotypes (*Emilsson et al., 2008*; *Albert and Kruglyak, 2015*; *Yao et al., 2015*; *GTEx Consortium et al., 2018*). In order to test the contribution of eQTLs to the response to stress, a phenotype that is likely to provide insight into disease, we overlapped our four response gene categories with genes whose expression level is associated with genetic variants in human heart tissues (eGenes; see Materials and methods). We observe a depletion of eGenes in the conserved response category, when compared to all expressed genes within each category, using data from the heart left ventricle (Chi-squared test; p=0.01), heart right atrial appendage (p=1.4×10$^{-5}$), and iPSC-derived cardiomyocytes (p=6.4×10$^{-5}$; *Figure 7A* and *Figure 7—figure supplement 1*). iPSC-derived cardiomyocytes consist mainly of ventricular-like cells. We therefore focused on eQTLs identified in the heart left ventricle. The depletion of eGenes corresponds to a difference in the contribution of eGenes to the non-response and conserved response categories (p<10$^{-15}$; *Figure 7C*). This observation is further supported by the fact that the absolute effect size of eQTLs, measured by allelic fold change, is significantly lower in the conserved response category compared to the non-response category (p=0.001; *Figure 7B*). We confirmed that there is no correlation between the eQTL effect size, and gene expression level in either response category (*Figure 7—figure supplement 1B*). The pattern of depletion of eGenes among conserved response genes is also observed across 12 other tested tissues; however the magnitudes of the effect differ between tissues (*Figure 7—figure supplement 1C*). We found that human-specific response genes are also depleted of eGenes when we considered all eGenes identified in at least 1 of the 14 tested tissues (p=7.1×10$^{-5}$), while chimpanzee-specific response genes are neither enriched nor depleted. Conversely, when we compare heart left ventricle eGenes to all expressed genes, we find that eGenes are depleted for conserved response genes (p=0.02). We next considered human gene expression response effect sizes, independent of response classification, in eGenes and non-eGenes. In accordance with the aforementioned findings, there is a lower absolute log fold change in expression in response to hypoxia in eGenes compared to non-eGenes (p<0.002; *Figure 7—figure supplement 2*).

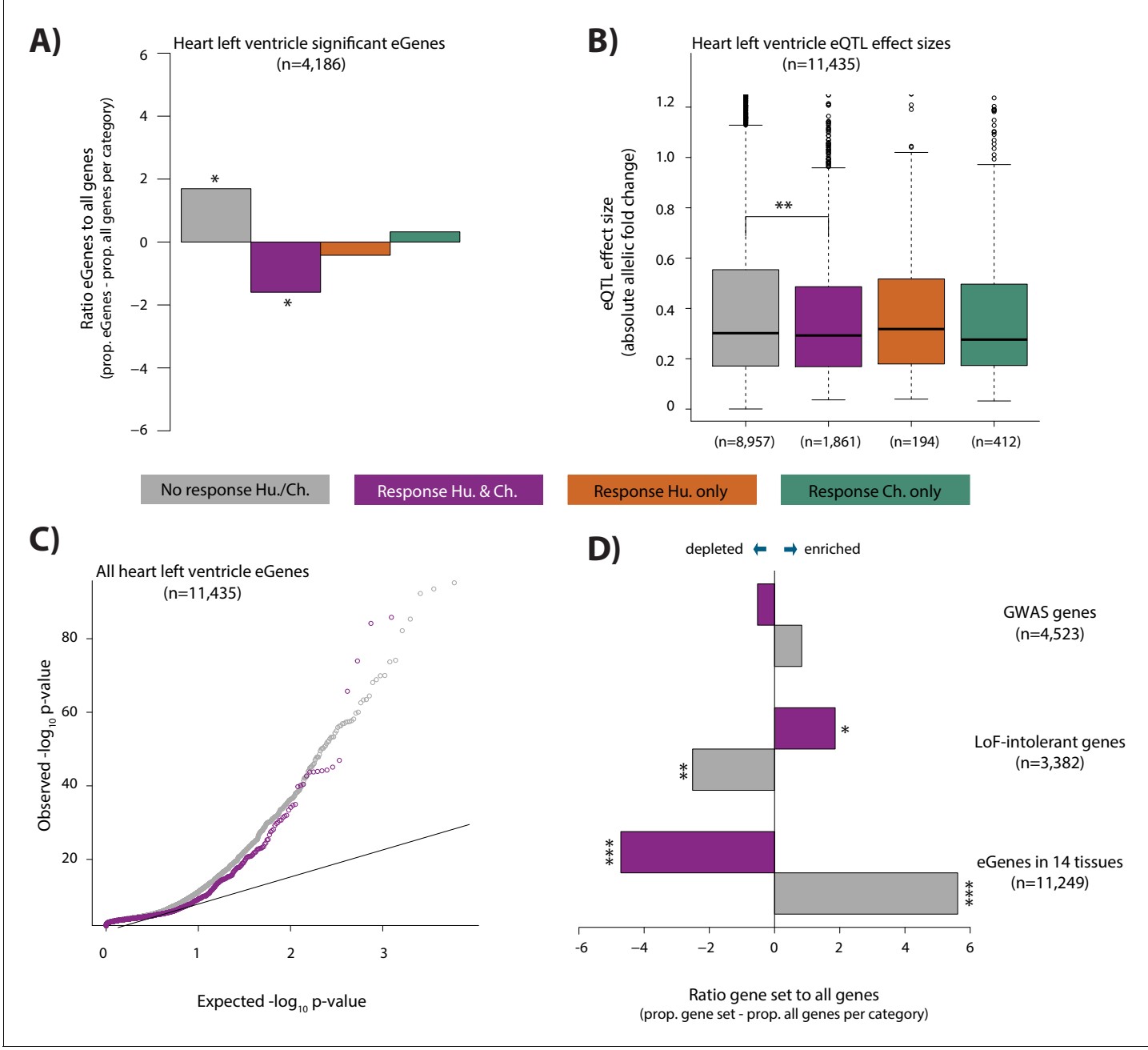

**Figure 7.** eGenes in heart tissue are depleted in conserved hypoxic response genes, while loss-of-function intolerant genes are enriched. (A) The proportion of heart left ventricle eGenes (GTEx Consortium) in each response category, relative to the proportion of all genes in each category. (B) eQTL effect size (defined as allelic fold change) of eGenes within the non-response, conserved response, human-specific, and chimpanzee-specific response categories. (C) QQ-plot representing all eGenes identified in heart left ventricle that overlap the non-response category (grey), and conserved response category (magenta). (D) The proportion of genes within three gene sets in the non-response and conserved response categories relative to the proportion of all genes within a response category. Gene sets include GWAS-associated genes (*Lek et al., 2016*), loss-of-function (LoF) intolerant genes (*Lek et al., 2016*), and eGenes identified in 14 tissues by the GTEx consortium. Also see *Figure 7—figure supplements 1–4*.

DOI: https://doi.org/10.7554/eLife.42374.025

The following figure supplements are available for figure 7:

**Figure supplement 1.** Conserved response genes are depleted for eGenes across tissues.

DOI: https://doi.org/10.7554/eLife.42374.026

**Figure supplement 2.** eGenes have lower gene expression response effect sizes between conditions, than non-eGenes.

DOI: https://doi.org/10.7554/eLife.42374.027

*Figure 7 continued on next page*

*Figure 7 continued*

**Figure supplement 3.** Conserved hypoxic response genes are depleted of ischemia eGenes, and enriched for ischemia response genes.
DOI: https://doi.org/10.7554/eLife.42374.028
**Figure supplement 4.** pLI scores of conserved response genes are higher than non-response genes.
DOI: https://doi.org/10.7554/eLife.42374.029

As we observed depletion of eQTLs found in healthy individuals among the conserved response genes in our study, we next considered eQTLs found among CVD patients. To do so, we investigated the contribution of eQTLs identified in left ventricle heart tissue from patients undergoing aortic valve replacement surgery pre- and post-cardioplegic arrest and ischemia, to our hypoxia response categories (*Stone et al., 2019*). Again, we observed a depletion of eGenes in the conserved response category for both pre- and post-ischemia eGenes (p=0.03 and p=0.006 respectively; *Figure 7—figure supplement 3A*). Interestingly, when we considered genes that are differentially expressed between pre- and post-ischemia samples, we observed the opposite pattern to that of the eGenes (i.e. differentially expressed genes are enriched in the conserved response category, and depleted in the non-response category); however the effect is not significant (p=0.08 for non-response genes and p=0.09 for conserved response genes; *Figure 7—figure supplement 3B*; see Materials and methods).

Differences in the gene expression response to hypoxia in eGenes compared to all expressed genes might imply differences in genetic tolerance to stress. We therefore overlapped our conserved response genes and non-response genes with genes associated with different levels of tolerance to mutation. When considering a set of 3,382 genes designated as loss-of-function intolerant in humans (*Lek et al., 2016*), we found that there is an enrichment of these genes in the conserved response category (p=0.02, *Figure 7D*; see Materials and methods), and depletion in the non-response category (p=0.005). Similarly, the probabilities of loss-of-function intolerance (pLI) scores for all genes in the conserved response category are significantly higher than those of genes within the non-response category (p=$2\times10^{-5}$, *Figure 7—figure supplement 4*). We then reasoned that 4,523 GWAS-associated genes are likely to be somewhat more tolerant to mutation than loss-of-function intolerant genes. Indeed, we found that there is no difference in the enrichment of these genes between the conserved response and non-response categories (*Figure 7D*).

## Cellular responses to hypoxia

We attempted to gain additional comparative insight in our system by characterizing cellular phenotypes that might relate to disease. First, we determined sensitivity to oxygen deprivation by measuring the level of cytotoxicity during the course of the experiment. A hallmark of cellular toxicity is the permeabilisation of the outer cellular membrane resulting in the release of intracellular components into the surrounding milieu. The activity of the lactate dehydrogenase (LDH) enzyme, which interconverts pyruvate and lactate, can be measured in the cell culture media as a proxy of this process. We observed a marginal yet significant increase in LDH activity following hypoxia in humans, and a significant increase following short-term re-oxygenation in both species (Student's t-test; p<0.05; *Figure 8A*). A significant increase is only observed in humans following hypoxia, and long-term re-oxygenation (human $B_{mean}$ = 0.61, human $D_{mean}$ = 4.82, chimpanzee $B_{mean}$ = 0.24, chimpanzee $D_{mean}$ = 2.49; Student's t-test; p<0.05 for human A vs. B and B vs. D). Despite these apparent within-species differences in response to hypoxia, there is no significant difference in LDH activity between species within a condition.

Second, we asked whether oxygen deprivation experienced by cardiomyocytes results in the secretion of cytokine signaling molecules, which could imply downstream consequences on other cell types in the heart such as cardiac fibroblasts. The TGFβ−1 cytokine mediates the development of fibrosis following stress in the heart (*Liu et al., 2017*). We therefore measured secreted TGFβ−1 by ELISA for four individuals in each species. We found that TGFβ−1 release was significantly increased following re-oxygenation after hypoxic stress in both species (p<$7\times10^{-3}$); however there is no difference between species under any condition (*Figure 8B*). The results from these cellular assays support the gene expression data indicating a generally conserved response to hypoxia across species.

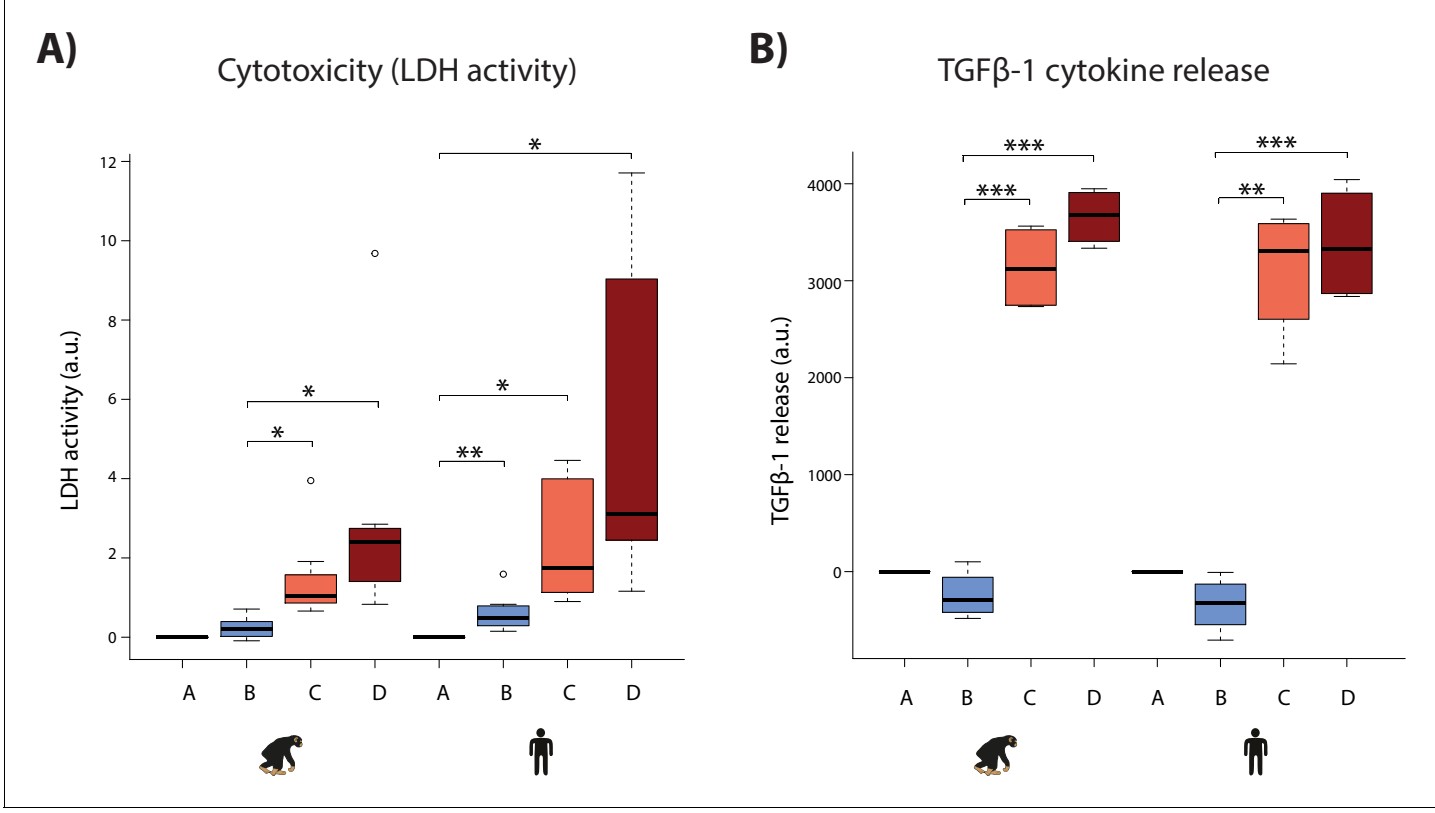

**Figure 8.** Hypoxia induces cytotoxicity and cytokine release in both species. (**A**) Levels of LDH activity, a measure of cytotoxicity, in the cell culture media from each phase of the experiment in all samples from each species. (**B**) Levels of TGFβ−1, a pro-fibrotic cytokine, in the cell culture media from four representative individuals of each species. Values are normalized to normoxia or hypoxia values that is A (A-A), B (B-A), C (C-B), and D (D-B). Values from the baseline normoxic (A: brown), hypoxic (B: blue), short-term re-oxygenation (C: coral), and long-term re-oxygenation (D: dark red) conditions are shown in each species. Asterisk denotes a significant difference between conditions (*p<0.05, **p<0.005, ***p<0.0005).
DOI: https://doi.org/10.7554/eLife.42374.030

## Discussion

Studying the human response to stress in an evolutionary context can potentially provide insight into disease susceptibility, incidence, aetiology, and response to treatment. In order to understand human adaptation and susceptibility to oxygen deprivation in the heart, we developed a cell-culture model to study the response to, and recovery from, hypoxia in iPSC-CMs from humans and chimpanzees. Using this system we were able to control exposure to changing oxygen levels in both species and study the ensuing in vitro response.

We found that, in humans and chimpanzees, the expression of ~4,000 genes (about a third of all expressed genes) is altered following six hours of hypoxic stress. Many of these genes return to baseline expression levels within 24 hr of re-oxygenation. The response to hypoxia is highly conserved in the two species with 1,920 genes responding similarly in humans and chimpanzees (75% of all genes that respond in at least one species). There have not been many comparative studies of functional perturbations in primates to provide us with broad context, but the conservation in response to hypoxia in our study is much greater than the conservation in immune response to infection between humans and chimpanzees (*Barreiro et al., 2010*). This suggests that the response to oxygen is a fundamentally conserved process across species, unlike the rapidly-evolving response to pathogens.

## Considering our observations in a broader context results in better functional insight

Conserved hypoxic response genes correspond to signaling pathways related to oxidative stress and hypoxia including the FOXO1 and HIF1 signaling pathways. However, genes responding to hypoxia are significantly depleted for known cardiovascular-associated genes, suggesting that hypoxic stress response genes are expressed, and active in multiple tissues. This is supported by the fact that we observe that conserved response genes are depleted for eGenes identified in the heart as well as in other tissues. These results suggest that there is less tolerance for genetic variability that results in variation in the expression of genes that are necessary for eliciting a response to stress. Indeed, while eGenes are depleted in conserved response genes, genes identified to be intolerant to loss of function, are enriched in conserved response genes. GWAS-associated genes, likely to have individually small effects on a trait, and therefore more tolerance to mutation, are neither enriched nor depleted in conserved response genes. Cellular stress, including oxidative stress, can contribute to cellular damage and lead to disease pathology (*Giordano, 2005*; *Sack et al., 2017*).

A common notion is that genetic variants that modulate gene expression levels (eQTLs) are important in mediating disease phenotypes (*Emilsson et al., 2008*; *Albert and Kruglyak, 2015*; *Yao et al., 2015*; *GTEx Consortium et al., 2018*). Thousands of genetic variants that associate with various phenotypes, including disease presentation, have been identified through genome-wide association studies (GWAS). Given that most of these variants are located within non-coding regions of the genome, it is thought that integrating GWAS data with eQTL data will help to identify genes that are relevant to the phenotype of interest (*Hormozdiari et al., 2016*; *Zhu et al., 2016*). The observation that up to half of GWAS-identified variants are also eQTLs in at least one tissue (*Battle et al., 2017*), provided some measure of support for this notion. However, there are several lines of evidence, which indicate that the relationship between eQTLs and complex disease may be relevant mainly to diseases that manifest late in life, and are therefore unlikely to have an effect on fitness: (i) Highly constrained genes with missense and protein-truncating variants, that might be expected to contribute to disease, are depleted for eQTLs but enriched for GWAS variants (*Lek et al., 2016*). (ii) Most eQTLs are shared across a large number of tissues and are expected to have broad functional effects, which are therefore unlikely to be highly deleterious (*Battle et al., 2017*). (iii) eQTLs identified in other primates are often also eQTLs in human (*Tung et al., 2015*; *Jasinska et al., 2017*) suggesting that, across species, eGenes can tolerate the accumulation of associated mutations, which perturb their regulation. Together, these findings suggest that most eQTLs may be neutral. In this study, we performed perturbation experiments to determine the consequences on gene expression across species. Our results, which show a depletion of eQTLs in genes that respond to hypoxic stress in cardiomyocytes across species, suggest that eQTLs alone may not give immediate insight into stress phenotypes associated with cardiovascular disease.

Genetic variants that modulate gene expression levels only in response to direct perturbation (response QTLs) are more likely to be informative in disease. Indeed, the association between GWAS variants is more pronounced in response QTLs than naïve QTLs (*Barreiro et al., 2012*; *Alasoo et al., 2018*). There are currently a limited number of data sets that allow for a systematic investigation of this association. However, using an available data set of heart tissue from CVD patients, we again observed a depletion of eQTLs in stress response genes. Our results suggest that eQTLs correspond to a set of genes that are largely distinct from genes, which respond to stress, or which are relevant to disease. Indeed, genes that are differentially expressed between pre- and post-ischemia samples show the opposite pattern to that of eQTLs in our data that is they are enriched in genes that respond to stress across species. While many eQTLs act in *cis*, it has been suggested that *trans*-eQTLs are more likely to associate with complex traits (*Westra et al., 2013*; *Battle et al., 2017*). However, we are currently underpowered to confidently identify these variants and determine their relevance to stress and disease.

In addition to quantifying gene expression levels in response to hypoxia, we measured cellular stress phenotypes in our comparative cardiomyocyte system. The baseline level of DNA oxidation damage is similar in humans and chimpanzees, and increases following recovery from hypoxia in both species. The baseline level of lipid peroxidation is also similar in humans and chimpanzees, and shows a trend towards increased levels during the course of the experiment in both species; however this increase is only significant in chimpanzees. These findings are in line with a study of

oxidative stress markers in blood from ten male humans and ten male chimpanzees, which showed that there is no significant difference in the levels of 8-OHdG between species, but there is significantly elevated 8-iso-PGF2α levels in chimpanzees compared to humans (*Videan et al., 2009*). Cytoxicity and cytokine release also increase during the course of the experiment in both species. Another comparative study on the effects of hypoxia in human and rhesus macaque cardiomyocytes demonstrates that the secreted metabolome is highly correlated between species after 24 hr of hypoxia (*Zhao et al., 2018*), suggesting an additional layer of regulatory conservation.

## Inter-species differences in response to oxygen perturbation

Although the overall correlation in the response to hypoxia is high between humans and chimpanzees, a stringent interaction analysis identified 147 genes with species-specific expression in the hypoxic condition.

The most significant species-specific response gene is *RASD1,* which is similarly expressed in humans and chimpanzees in normoxic conditions but is significantly up-regulated after hypoxia only in humans. *RASD1* was found to be up-regulated in samples from patients with ischemic disease compared to patients with dilated cardiomyopathy (heart damage despite normal blood flow), and non-failing hearts (*Liu et al., 2015*). These results suggest that aberrant *RASD1* expression could be specifically involved in the response to oxygen deprivation, and the pathogenesis of ischemic heart disease.

The *RAI1-PEMT-RASD1* region is a replicated, genome-wide significant locus for coronary artery disease (CAD) (*McPherson and Tybjaerg-Hansen, 2016*). It is unclear what pathway, related to the CAD phenotype, is affected by the *RASD1* locus (*Khera and Kathiresan, 2017*), and further experimentation specifically focused on this gene is beyond the scope of the current study. That said, several previous observations are also consistent with the notion of a relationship between *RASD1* expression and the response to oxygen. The SNP in the *RASD1-PEMT-RAI* locus that is associated with CAD is not associated with various vascular-related traits (*Schunkert et al., 2011*); yet it is associated with ischemic stroke, with the same direction of effect (*Dichgans et al., 2014*), suggesting a potential role for oxygen deprivation. It was also previously reported that many CAD case participants in GWAS studies have suffered from myocardial infarction, which results in myocardial ischemia and hypoxia, and that there is evidence for the SNP-GWAS association in the myocardial infarction sub-phenotype of CAD (*Schunkert et al., 2011*). Moreover, the SNP is an eQTL for *RASD1* expression in monocytes (*Emilsson et al., 2008*; *Schunkert et al., 2011*), suggesting that it has the potential to influence *RASD1* expression in the right context. Finally, while *RASD1* expression is induced following hypoxia in humans only, the *RAI1* gene within this locus responds to hypoxia in both species, suggesting a link between hypoxia and this locus (*PEMT* expression doesn't significantly change in either species upon hypoxia). In addition to *RASD1*, there are four other CAD-associated genes (*MRPS6, SWAP70, SNF8* and *TRIB1*) that respond to hypoxia in a species-specific manner. *MRPS6*, another human-specific response gene, encodes a mitochondrial ribosomal protein likely important in the translation of mitochondrial mRNAs necessary for oxidative phosphorylation.

RASD1 is an activator of G-protein signaling (*Cismowski et al., 2000*), and is thought to contribute to the stress response (*Sato and Ishikawa, 2010*). Indeed, G-proteins are important sensors of the environment at the cell membrane and mediate a signaling cascade to initiate an intra-cellular response to external stimuli. *RASD1* is one of several species-specific response genes related to G-protein signaling; other examples include *ARRDC2, RASL11B, ARL6 RASSF1, GTPBP4, RAB3A, RHOF, SYDE2,* and *CNKSR1*.

In fact, there are several genes that respond in a species-specific manner, which belong to similar pathways, or perform similar functions. For example, multiple genes (*ACVR2A, SNIP1, JUNB, SMAD4, TGFBR2, SMAD6, FGF9*) are related to TGF-β signaling. TGF-β is induced following myocardial infarction, and mediates the development of fibrosis (*Bujak and Frangogiannis, 2007*). While we did not observe differences in the secretion of TGFβ−1, it is tempting to speculate that differences in the expression of these genes under oxygen stress, could contribute to the fibrotic heart phenotype observed in chimpanzees. Several of these genes have been implicated in heart physiology and disease suggesting that they could be relevant to the phenotype (*Galvin et al., 2000*; *Wang et al., 2005*; *Alfonso-Jaume et al., 2006*; *Tseng et al., 2009*; *Itoh et al., 2016*; *Lu et al., 2016*; *Dogra et al., 2017*).

Many species-specific response genes are involved in post-transcriptional layers of the gene regulatory cascade including RNA modifications (*METTL14*), RNA folding (*DDX20*), splicing (*CCDC49*), nuclear-cytoplasmic transport (*NUP214*), and protein degradation (*CBLL1, UBQLN4, TRIM13, DNAJB2*). This suggests that additional inter-species differences may emerge in processes downstream of transcription. For example, METTL14 deposits the N$^6$-methyladenosine RNA modification, which has been implicated in the stabilization of mRNA molecules following hypoxia (*Fry et al., 2017*). Intriguingly, differences in N$^6$-methyladenosine levels have been reported between primates (*Ma et al., 2017*). Alternative splicing is another mechanism for inter-species differences between humans and chimpanzees, and, interestingly, one gene that undergoes differential splicing between species is the *GSTO2* gene, which is protective against oxidative stress (*Calarco et al., 2007*).

Genes that respond to oxygen deprivation in a species-specific manner could have phenotypic consequences as rapid changes in gene expression in response to stress can lead to evolutionary adaptation (*López-Maury et al., 2008*; *de Nadal et al., 2011*). This mechanism has been implicated in mediating inter-species differences in epithelial cancer incidence as coordinated gene expression differences between humans and chimpanzees have been observed in fibroblasts subjected to serum starvation (*Pizzollo et al., 2018*). Conversely, it has been suggested that species-specific gene responses to stress across divergent yeast species may be adaptive, or, more likely, compensated by the response of related genes, or reflective of biological noise (*Tirosh et al., 2011*). The latter could explain the overall inter-species similarity in the cellular and transcriptional response, and the fact that genes peripherally related to similar pathways show species-specific differences.

## Potential limitations of our model

We performed our experiments using an in vitro iPSC-CM model. To characterize the potential relevance of our observations to in vivo systems, we considered the results of Pavlovic *et al.*, who characterized regulatory differences between primary heart tissues and iPSC-CMs in humans and chimpanzees (*Pavlovic et al., 2018*). Out of 2,459 genes that respond to hypoxia in one or both species based on our data, only 371 (16%) were found to be differentially expressed between hearts and iPSC-CMs in Pavlovic *et al*. Similarly, 34 of the 147 species-by-condition interaction genes in our data (23%) were found to be differentially expressed between hearts and iPSC-CMs. In addition, we find that there is no significant difference in the proportions of conserved response genes and non-response genes in the overlap with genes that are differentially expressed between heart tissue and iPSC-CMs. Similarly, there is no significant difference in the proportion of conserved response and species-specific response genes that overlap with genes differentially expressed between heart tissue and iPSC-CMs. Put together, this analysis suggests that there is no systematic bias in our cell culture system.

It is important to note that following myocardial infarction and ischemia, highly metabolically active cardiomyocytes undergo rapid cell death thereby initiating a cascade of events including an inflammatory response by immune cells, cardiac fibroblast activation, and fibrosis (*Frangogiannis, 2014*). Our study was designed to measure the primary response to oxygen deprivation in the heart at the level of cardiomyocytes. Given the high mitochondrial content of cardiomyocytes, these cells are likely to be more susceptible to oxidative stress than other cell types. Indeed, cardiomyocytes are more sensitive to superoxide radicals than cardiac fibroblasts (*Li et al., 1999*). However, our system was not able to capture secondary effects on extracellular matrix remodeling and fibroblast proliferation (*Ugolini et al., 2017*), which may differ between species and also contribute to the different disease phenotypes. We attempted to measure secondary disease processes by assaying TGFβ−1 secretion by cardiomyocytes, but did not find a significant difference in the release of this factor by cardiomyocytes between species.

In summary, to date there have been few well-powered studies investigating the evolution of the stress response in primates. Here we measured the genome-wide transcriptional response to a universal cellular stress, oxygen deprivation, across species in a CVD-relevant cell type. We find that the cellular and transcriptional response is largely similar across species; however there are hundreds of genes that respond in a species-specific manner.

# Materials and methods

**Key resources table**

| Reagent type (species) or resource | Designation | Source or reference | Identifiers | Additional information |
|---|---|---|---|---|
| Cell line (H.sapiens, Female) | H20682 iPSC | *Ward et al., 2018* | | |
| Cell line (H.sapiens, Male) | H20961 iPSC | *Burrows et al., 2016* | | |
| Cell line (H.sapiens, Female) | H21792 iPSC | *Ward et al., 2018* | | |
| Cell line (H.sapiens, Female) | H22422 iPSC | This study | | Age 19, Caucasian, fibroblast origin |
| Cell line (H.sapiens, Female) | H24280 iPSC | *Pavlovic et al., 2018* | | |
| Cell line (H.sapiens, Female) | H25237 iPSC | This study | | Age unknown, Caucasian, fibroblast origin |
| Cell line (H.sapiens, Male) | H28126 iPSC | *Burrows et al., 2016* | | |
| Cell line (H.sapiens, Male) | H28815 iPSC | *Ward et al., 2018* | | |
| Cell line (P.troglodytes, Female) | C3647 iPSC | *Gallego Romero et al., 2015* | | |
| Cell line (P.troglodytes, Male) | C3649 iPSC | *Gallego Romero et al., 2015* | | |
| Cell line (P.troglodytes, Female) | C40210 iPSC | *Gallego Romero et al., 2015* | | |
| Cell line (P.troglodytes, Female) | C40280 iPSC | *Gallego Romero et al., 2015* | | |
| Cell line (P.troglodytes, Female) | C40300 iPSC | *Pavlovic et al., 2018* | | |
| Cell line (P.troglodytes, Male) | C4955 iPSC | *Gallego Romero et al., 2015* | | |
| Cell line (P.troglodytes, Male) | C8861 iPSC | *Gallego Romero et al., 2015* | | |

## Samples

We used eight biological replicates (individuals) from human, and seven from chimpanzee. In addition, technical replicates (independent cardiomyocyte differentiation and oxygen stress experiments) from three human and three chimpanzee individuals were used to estimate unwanted factors of variation in the data. This number of biological and technical replicates is sufficient to be able to identify inter-species gene expression differences (*Gallego Romero et al., 2015*; *Pavlovic et al., 2018*; *Ward et al., 2018*). All iPSC lines, from both species, were derived from fibroblasts using the same experimental design and reprogramming protocol as previously described (*Gallego Romero et al., 2015*). Regardless, Gallego Romero *et al.* found the effects of different reprogramming protocols, population of origin, and originating cell types on inter-species DNA methylation and gene expression differences to be exceedingly small (*Gallego Romero et al., 2015*). 13 iPSC lines have been described and characterized previously (*Gallego Romero et al., 2015*; *Burrows et al., 2016*; *Pavlovic et al., 2018*; *Ward et al., 2018*). Two additional iPSC lines are first described and characterized in this study (H22422 and H25237). All lines tested negative for mycoplasma contamination.

## Differentiating cardiomyocytes from iPSCs

Feeder-independent iPSCs were maintained at 70% confluence on Matrigel hESC-qualified Matrix (354277, Corning, Bedford, MA, USA) at a 1:100 dilution. Cells were cultured in Essential 8 Medium (A1517001, ThermoFisher Scientific, Waltham, MA, USA) with Penicillin/Streptomycin (30002 Cl,

Corning) at 37°C, 5% $CO_2$ and atmospheric $O_2$. Cells were passaged every 3–4 days with dissociation reagent (0.5 mM EDTA, 300 mM NaCl in PBS), and seeded with ROCK inhibitor Y-27632 (ab12019, Abcam, Cambridge, MA, USA). iPSC-CM differentiations were largely performed based on the protocol described by *Burridge et al. (2014)*. Importantly, the same differentiation protocol was used in both species. iPSCs cultured for 10–50 passages were seeded in 4 × 10 cm Matrigel-coated culture dishes until 70–100% confluent (Days −4/–3). The optimum cell density for efficient differentiation depended on the individual iPSC line. On Day 0, 6 µM of the GSK3 inhibitor, CHIR99021 trihydrochloride (4953, Tocris Bioscience, Bristol, UK) was added to the cultures in 12 ml Cardiomyocyte Differentiation Media [500 mL RPMI1640 (15–040 CM ThermoFisher Scientific), 10 mL B-27 Minus Insulin (A1895601, ThermoFisher Scientific), 5 mL Glutamax (35050–061, ThermoFisher Scientific), and 5 mL Penicillin/Streptomycin)], and a 1:100 dilution of Matrigel. 24 hr later, on Day 1, fresh Cardiomyocyte Differentiation Media, supplemented with 6 µM CHIR99021 was added to the cultures. Chimpanzee iPSCs, in general, were more sensitive to the addition of CHIR99021 hence the reduction from the optimal 12 µM CHIR99021 for 24 hr as described in Burridge *et al.*, to 6 µM for 48 hr. The GSK3 inhibitor was removed with the addition of fresh Cardiomyocyte Differentiation Media 24 hr later on Day 2. After 24 hr, on Day 3, 2 µM of the Wnt signaling inhibitor Wnt-C59 (5148, Tocris Bioscience), diluted in Cardiomyocyte Differentiation Media, was added to the cultures. The media was replaced with 2 µM Wnt-C59 in Cardiomyocyte Differentiation Media 24 hr later, on Day 4. Cardiomyocyte Differentiation Media was replaced on Days 5, 7, 10 and 12. Spontaneously beating cells appear on Days 7–10.

To remove non-cardiomyocytes from the cultures, iPSC-CMs were purified by metabolic selection. 10 mL of glucose-free, lactate-containing media (Purification Media) [500 mL RPMI without glucose (11879, ThermoFisher Scientific), 106.5 mg L-Ascorbic acid 2-phosphate sesquimagenesium salt (sc228390, Santa Cruz Biotechnology, Santa Cruz, CA, USA), 3.33 ml 75 mg/ml Human Recombinant Albumin (A0237, Sigma-Aldrich, St Louis, MO, USA), 2.5 mL 1 M lactate in 1 M HEPES (L(+)Lactic acid sodium (L7022, Sigma-Aldrich)), and 5 ml Penicillin/Streptomycin] was added on Day 14. Purification Media was replaced on Days 16 and 18. On Day 20, iPSC-CMs were dissociated with 4 mL 0.05% Trypsin-EDTA solution (25–053 Cl, ThermoFisher Scientific) for ~10 min, and quenched with double the volume of Cardiomyocyte Maintenance Media [500 mL DMEM without glucose (A14430-01, ThermoFisher Scientific), 50 mL FBS (S1200-500, Genemate), 990 mg Galactose (G5388, Sigma-Aldrich), 5 mL 100 mM sodium pyruvate (11360–070, ThermoFisher Scientific), 2.5 mL 1 M HEPES (SH3023701, ThermoFisher Scientific), 5 mL Glutamax (35050–061, ThermoFisher Scientific), 5 mL Penicillin/Streptomycin]. This change in carbohydrate source shifts metabolism away from glycolysis towards aerobic mitochondrial respiration, which is the predominant pathway used to generate energy in adult cardiomyocytes in vivo. A single cell suspension was generated by straining the cells through a 100 µm nylon mesh cell strainer three times, and once through a 40 µm mesh strainer. 1.5 million iPSC-CMs were plated per well of a Matrigel-coated 6-well plate in 3 mL Cardiomyocyte Maintenance Media. iPSC-CMs for each of the four conditions were plated on separate 6-well plates.

## Culturing iPSC-CMs

iPSC-CMs were matured in culture for a further 10 days. Cardiomyocyte Maintenance Media was replaced on Days 23, 25, 27, 28 and 30.

While cells are typically cultured in vitro at atmospheric oxygen levels (21% $O_2$), this oxygen level is not experienced by mammalian cells in vivo - arterial oxygen levels are ~13%, and levels drop to 5–10% within tissues (*Brahimi-Horn and Pouysségur, 2007*; *Carreau et al., 2011*; *Jagannathan et al., 2016*). We therefore chose to culture our cells at 10% $O_2$, which falls within this physiological range. On Day 25, iPSC-CMs cultured at atmospheric oxygen levels, were transferred to an oxygen-controlled incubator (HERAcell 150i $CO_2$ incubator, ThermoFisher Scientific) representing physiological oxygen levels (10% $O_2$). Oxygen levels are maintained through displacement of oxygen by nitrogen.

To allow further maturation, and to synchronize iPSC-CM beating, iPSC-CMs were pulsed with the IonOptix C-Dish and C-Pace EP Culture Pacer from Day 27 until the end of the oxygen perturbation experiment. Cells were pulsed at a voltage of 6.6 V/cm, frequency of 1 Hz, and pulse frequency of 2 ms.

## Immunocytochemistry iPSC-CM samples were dual-stained in two independent staining reactions

### Stain 1

Cells were fixed with 3–4% paraformaldehyde for 15 min at room temperature and then washed three times with PBS. Cells were permeabilized with 0.3% Triton X-100 for 10 min prior to another three washes with PBS. Cells were blocked with 5% BSA for 30 min and then incubated with primary antibodies diluted in 5% BSA O/N at 4˚C. Anti-Cardiac Troponin T rabbit polyclonal antibody (ab45932, Abcam) was added at a 1:400 dilution. Anti-alpha-Actinin (Sarcomeric) mouse monoclonal antibody (A7811, Sigma) was added at a 1:500 dilution. After primary antibody incubation, cells were washed with 0.1% Tween-20 in PBS three times. Secondary antibodies, donkey anti-Rabbit IgG Alexa Fluor 594 (A21207, ThermoFisher Scientific), and Donkey anti-Mouse IgG Alexa Fluor 488 (A21202, ThermoFisher Scientific), were added at a 1:1000 dilution in 2.5% BSA for 1 hr at room temperature. Cells were washed three times with PBS prior to counter-staining with Hoechst for 10 min.

### Stain 2

Cells were fixed with 3–4% paraformaldehyde for 15 min at room temperature and washed three times with PBS. Cells were not permeabilized or blocked. Primary antibodies were diluted in Permeabilization Buffer (FOXP3/Transcription Factor Staining Buffer Set, 00–5523, ThermoFisher Scientific), and incubated O/N at 4˚C. Anti-IRX4 rabbit polyclonal antibody (ab123542, Abcam) at a 1:200 dilution, and Cardiac Troponin T Monoclonal Antibody (MA5-12960 clone 13–11, ThermoFisher Scientific) at a 1:200 dilution, were used. Cells were washed three times with Permeabilization Buffer after primary antibody incubation. Secondary antibodies were added at a 1:1000 dilution in Permeabilization Buffer for 1 hr at room temperature. Donkey anti-Mouse IgG Alexa 594 (A21203, ThermoFisher Scientific), and Donkey anti-Rabbit IgG Fluor 488 (A21206, ThermoFisher Scientific), were used. Cells were washed three times with PBS prior to counter-staining with Hoechst for 10 min.

## Flow cytometry

iPSC-CMs were dissociated with 0.05% Trypsin-EDTA solution and quenched with four times the volume of Cardiomyocyte Maintenance Media. In order to obtain a single cell suspension, cells were strained twice through a 100 µm strainer, and once through a 40 µm strainer. 1 million cells were stained with Zombie Violet Fixable Viability Kit (423113, BioLegend) for 30 min at 4˚C prior to fixation and permeabilization (FOXP3/Transcription Factor Staining Buffer Set, 00–5523, ThermoFisher Scientific) for 30 min at 4˚C. Cells were stained with 5 µl PE Mouse Anti-Cardiac Troponin T antibody (564767, clone 13–11, BD Biosciences, San Jose, CA, USA) for 45 min at 4˚C. Cells were washed three times in permeabilization buffer and re-suspended in autoMACS Running Buffer (130-091-221, Miltenyi Biotec, Bergisch Gladbach, Germany). Several negative controls were used in each flow cytometry experiment: 1) iPSCs, which should not express TNNT2, 2) an iPSC-CM sample that has not been labeled with viability stain or TNNT2 antibody, and 3) an iPSC-CM sample that is only labeled with the viability stain.

10,000 cells were captured and profiled on the BD LSRFortessa Cell Analyzer. Several gating steps were performed to determine the proportion of TNNT2-positive cells: 1) Cellular debris was removed by gating out cells with low granularity on FSC versus SSC density plots, 2) From this population, live cells were identified as the violet laser-excitable, Pacific Blue dye-negative population, 3) Two populations of TNNT2-positive cells were identified within the set of live cells: one conservative gate selected high-intensity TNNT2-positive cells, and a lenient gate included TNNT2-positive cells with a range of intensities. Both gates were created so as to exclude any cells that overlap the profiles of the negative control samples. iPSC-CM purity is reported as the proportion of TNNT2-positive live cells. Values from the lenient threshold are reported in the main text. Values for the conservative and lenient thresholds are reported in *Figure 1—figure supplement 3* and *Supplementary file 1*-Table S2. Importantly, there is no difference in purity between species whether a conservative or lenient threshold is used.

## Oxygen stress experiment

Several pilot experiments were performed to determine the optimal oxygen conditions to initiate a gene expression response in iPSC-CMs. The expression of stress response genes is induced within 6 hr of hypoxia at 1% oxygen (*Figure 1—figure supplement 4B*), and cell damage (lactate dehydrogenase activity) is evident after 24 hr (*Figure 1—figure supplement 4C*). It is noteworthy that culturing the iPSC-CMs in media with a glucose carbohydrate source, instead of galactose, does not elicit a stress response following hypoxia (*Figure 1—figure supplement 4*). Given that we were interested in determining the early transcriptional response to hypoxia, prior to the induction of cell death, we chose the 6 hr time-point in our subsequent experiments.

Oxygen stress experiments were conducted on Day 31 or 32 after the initiation of differentiation. At the start of the experiment (total elapsed time = 0 hr), one plate (A) remained at 10% $O_2$, while three plates (B, C and D) were transferred to an oxygen-controlled cell culture incubator set at 1% $O_2$. After six hours, plates A and B were harvested, while plates C and D were transferred back to 10% $O_2$ (elapsed time = 6 hr). Plate C was harvested six hours after the end of the hypoxic incubation (elapsed time = 12 hr). Plate D was harvested 24 hr after the end of the hypoxic incubation (elapsed time = 30 hr). Ten batches of oxygen stress experiments were performed. For each of the four conditions, iPSC-CMs were harvested by manual scraping, flash-frozen as cell pellets, and stored at −80°C, together with the cell culture media from each sample, until further processing.

Oxygen levels in the cell culture media of a representative iPSC-CM sample were measured during the course of the oxygen perturbation experiment, in each experimental batch. An oxygen sensitive sensor was applied to the inner wall of a well of a 6-well plate (SP-PSt3-NAU-D5-YOP, PreSens Precision Sensing GmbH, Regensburg, Germany). Oxygen levels were measured non-invasively through the wall of the cell culture plate using a Polymer Optical Fiber (NWDV29, Coy, Grass Lake, MI, USA), and a Fiber Optic Oxygen Meter (Fibox 3 Transmitter NWDV16, Coy).

## Oxidative DNA damage assay

8-OHdG levels were measured by competitive enzyme-linked immunoassay using the OxiSelect Oxidative DNA Damage ELISA Kit (STA-320, Cell Biolabs Inc). Levels were measured in 50 µl of cell culture media, in duplicate, according to the manufacturer's instructions. Samples were processed on three species-balanced 96-well plates. 8-OHdG was quantified relative to a standard curve using 4- and 5-parameter logistic models implemented in the drc package in R. Final 8-OHdG release is reported as four measurements either relative to the basline condition A or hypoxic condition B: A (A-A), B (B-A), C (C-B) and D (D-B). For the three individuals with replicate experiments in each species, mean values from both experiments are reported.

## Lipid peroxidation assay

Secreted 8-iso-PGF2α was measured by competitive enzyme-linked immunoassay using the OxiSelect 8-iso-Prostaglandin F2α ELISA kit (STA-337, Cell Biolabs Inc, San Diego, CA, USA). Levels were measured in 55 µl of cell culture media, in duplicate, according to the manufacturer's instructions. Samples were processed on three species-balanced 96-well plates. 8-iso-PGF2α was quantified relative to a standard curve using 4- and 5-parameter logistic models implemented in the drc package in R. Final 8-iso-PGF2α release is reported as four measurements either relative to the baseline, condition A, or the hypoxic condition that is A (A-A), B (B-A), C (C-B) and D (D-B). For the three individuals with replicate experiments in each species, mean values from both experiments are reported.

## Lactate dehydrogenase activity assay

Lactate dehydrogenase (LDH) activity levels were measured by colourimetric determination of NAD reduction to NADH using the Lactate Dehydrogenase Activity Assay Kit (MAK066, Sigma-Aldrich). Samples were processed on four species-balanced 96-well plates, and each sample was assayed in triplicate. 5 µl of cell culture media was assayed as per the manufacturer's instructions. LDH activity is reported as the difference in NADH levels measured at the start of the enzymatic reaction, and 25 min after the addition of the substrate. Enzyme activity is calculated relative to a linear standard curve. Final LDH activity is reported as four measurements either relative to the baseline, condition A, or the hypoxic condition that is A (A-A), B (B-A), C (C-B) and D (D-B). For the three individuals with replicate experiments in each species, mean values from both experiments are reported.

## TGFβ−1 ELISA assay

TGFβ−1 levels were measured by enzyme-linked immunoassay using the TGF beta 1 Human ELISA Kit (ab100647, Abcam). Levels were measured in 100 µl of cell culture media, in duplicate, according to the manufacturer's instructions. Four representative individuals from each species were assayed on one 96-well plate. TGFβ−1 levels were quantified relative to a standard curve using 4- and 5-parameter logistic models implemented in the drc package in R. Final TGFβ−1 release is reported as four measurements: A (A-A), B (B-A), C (C-B) and D (D-B).

## RNA-seq library preparation and sequencing

RNA was extracted from ~1.5 million cells from 84 iPSC-CM samples representing 15 individuals. Extractions were performed in ten species-balanced batches using the ZR-Duet DNA/RNA extraction kit (D7001, Zymo, Irvine, CA, USA). All four conditions from one human and one chimpanzee individual were extracted per batch (one batch had three individuals). RNA concentration and quality was measured using the Agilent 2100 Bioanalyzer. RIN scores were greater than 7.5 for all samples (human median: 9.1, chimpanzee median: 9.2).

RNA-seq libraries were prepared from 250 ng of RNA in three species-balanced batches using the Illumina TruSeq RNA Sample Preparation Kit v2 (RS-122–2001 and −2002, Illumina). Libraries in each batch were multiplexed together to generate four pools for sequencing. Each pool was sequenced 50 base pairs, single-end on the HiSeq4000 according to the manufacturer's instructions. Pools 1,3,4 were sequenced on three lanes (24 samples per pool), and Pool two was sequenced on two lanes (12 samples in the pool).

## RNA-seq data processing

RNA-seq data quality was determined by FastQC (http://www.bioinformatics.babraham.ac.uk/projects/fastqc/). Sequencing adapters were trimmed, and sequencing reads from each species aligned to their respective genome (hg19 or panTro3) using TopHat2 (version 2.0.11) (*Kim et al., 2013*). The number of mapped sequencing reads is similar across species and conditions (median human A: 53,547,009, median human B: 40,872,328; median human C: 40,635,553, median human D: 42,041,844; median chimpanzee A: 42,258,525; median chimpanzee B: 33,054,882; median chimpanzee C: 28,792,150, median chimpanzee D: 36,903,485). In order to have a comparable set of genes from which to identify gene expression differences, we quantified gene expression levels at orthologous meta-exons from 30,030 Ensembl genes from hg19, panTro3 and rheMac3 (*Blekhman et al., 2010*) using featureCounts within subread (version 1.4.6) (*Liao et al., 2014*). In order to compare gene expression profiles equivalently across individuals regardless of sex, genes were filtered to only include those on the autosomes. $Log_2$-transformed counts per million were calculated using edgeR (*Robinson et al., 2010*). Lowly-expressed genes were filtered such that only genes with a mean $log_2$ cpm >0 across samples were retained.

Prior to differential expression analysis, unwanted factors of variation were estimated in the RNA-seq data using RUVSeq (*Risso et al., 2014*). As we have two replicate samples from six of the individuals (three in each species), the RUVs function for estimating the factors of unwanted variation using replicate samples was used. This approach takes advantage of the fact that replicate samples have constant covariates of interest. We tested different numbers of unwanted factors of variation (k values) until the data clustered best by our biological factors of interest that is species, individual and condition. Four factors of unwanted variation were thus selected.

To assess data quality we performed Principal Component Analysis (PCA) on the RUVs-normalised $log_2$ cpm expression values. We correlated known biological and technical factors with the first six PCs.

## Differential expression analysis

The TMM-voom-limma pipeline was used to identify differentially expressed genes between species and conditions. Filtered read counts from a randomly selected replicate were taken forward in this analysis. Normalization factors were used to scale the raw library size to the effective library size of each sample using the trimmed mean of M-values (TMM) implemented in edgeR (*Robinson et al., 2010*). The mean-variance relationship was removed using precision weights in voom (*Law et al., 2014*).

A linear model was fitted for expression values of each gene using limma (*Smyth, 2004*):

$$Y \sim \beta_0 + \beta_{species}X_{species} + \beta_B X_B + \beta_C X_C + \beta_D X_D + \beta_{species,B}X_{species,B} +$$
$$\beta_{species,C}X_{species,C} + \beta_{species,D}X_{species,D} + X_{RUV1} + X_{RUV2} + X_{RUV3} + X_{RUV4} + I + \epsilon$$

where $\beta_0$ is the mean expression level of gene g for chimpanzee cells grown under normoxic conditions (A), $\beta_{species}$ is the fixed effect for species, $\beta_B$ is the effect for condition B, $\beta_C$ is the effect for condition C, $\beta_D$ is the effect for condition D, $\beta_{species,B}$ is the fixed interaction effect of condition B and species, $\beta_{species,C}$ is the fixed interaction effect of condition C and species, $\beta_{species,D}$ is the fixed interaction effect of condition D and species. The four unwanted factors of variation determined by RUVs are modeled as covariates $X_{RUV1}$, $X_{RUV2}$, $X_{RUV3}$, $X_{RUV4}$, and $I$ is the random effect for individual, which was implemented using the limma function duplicateCorrelation.

In order to obtain more precise gene-wise variability estimates, we used empirical Bayes moderation, which takes information across all genes into account. We used contrast tests in limma to identify genes that are differentially expressed between conditions within each species, genes that are differentially expressed between species at each condition, and species-by-condition interactions for conditions B, C, and D. We corrected for multiple testing at each gene using the Benjamini and Hochberg false discovery rate (FDR) (*Benjamini and Hochberg, 1995*). Genes with FDR-adjusted p values of < 0.1 are considered to be differentially expressed.

## Identification of gene expression trajectories

In order to cluster genes by their gene expression trajectories during the course of the experiment, all data was jointly modeled using a Bayesian Hierarchical model across pairwise differential tests implemented in the Cormotif R package (*Wei et al., 2015*). Cormotif fits correlation motifs to multiple pairs of tests to identify differential expression patterns. To identify gene expression trajectories (correlation motifs), TMM-normalised cpm values for each gene were compared across three pairs of conditions within each species (6 pairs of tests in total; 1: human A vs. B, 2: human B vs. C, 3: human B vs. D, 4: chimpanzee A vs. B, 5: chimpanzee B vs. C, 6: chimpanzee B vs. D). In order to select the best model to fit the data, we varied the number of correlation motifs (1 through 15). The best fit was determined using the Bayesian information criterion (BIC) and Akaike information criterion (AIC). The BIC and AIC were minimized when four correlation motifs were modeled. Cormotif calculates the posterior probability of differential expression for each gene, in each of the six pairwise tests. These values are plotted in the heatmap in *Figure 5A*. We used a threshold posterior probability of 0.5 to classify genes into each of the four correlation motifs as suggested by the authors. Motif 1 (non-response): p<0.5 in tests 1,2,3,4,5,6; motif 2 (chimpanzee-specific response): p<0.5 in tests 1,2,3 and p>0.5 in tests 4,5,6; motif 3 (conserved response): posterior probability >0.5 in tests 1,2,3,4,5,6; motif 4 (human-specific response): p>0.5 in tests 1,2,3 and p<0.5 in tests 4,5,6.

## Comparison of our data to that collected in human and rhesus macaque iPSC-CMs

The set of 187 genes that were previously found to respond to hypoxia in both human and rhesus macaque iPSC-derived cardiomyocytes were overlaid with our four Cormotif gene expression response categories (*Zhao et al., 2018*). 164 of the human-rhesus conserved response genes are expressed in our data.

We calculated the proportion of human-rhesus conserved response genes within each of our four response gene categories (value one for each category), and the proportion of all genes that belong to each of the four response categories in our data (value two for each category). We subtracted value twofrom value one to obtain an enrichment score per response category. This score is multiplied by 100 and plotted in *Figure 6—figure supplement 1*. A chi-squared test is used to determine whether there is a significant enrichment or depletion of human-rhesus response genes in each of our four response gene categories compared to all expressed genes in that category. This approach is used in all subsequent analyses investigating the properties of the four response categories.

## Oxygen-sensitive transcription factor binding integration analysis

Human transcription start sites (TSS) were downloaded from the UCSC genome bowser Table Browser (http://genome.ucsc.edu/cgi-bin/hgTables) using 'txStart' from Ensembl genes

(*Karolchik, 2004*). Each gene was assigned a single TSS based on the 5' most transcript of genes on the sense strand, and the 3' most transcript from genes on the anti-sense strand. The list of TSS was filtered to include only those representative of orthologous genes used in the gene expression analysis.

We obtained published human ChIP-seq data sets for HIF1$\alpha$ (356 sites), HIF2$\alpha$ (301 sites) (*Schödel et al., 2011*), and FOXO3 (934 sites) (*Eijkelenboom et al., 2013*). HIF1$\alpha$ and HIF2$\alpha$ binding was measured in a breast cancer cell line following HIF stabilization with dimethyloxaloylglycine. FOXO3 binding was measured in a colon cancer cell line with a Tamoxifen-inducible FOXO3A3-ER fusion protein. Binding locations for each transcription factor were converted to hg19 coordinates. Each transcription factor binding location was assigned its closest gene using bedtools (*Quinlan and Hall, 2010*). Genes were stratified into each of the four response gene categories, and the proportion of genes within each class calculated as described previously. The conservation status of each transcription factor binding location was determined using the phyloP score (*Pollard et al., 2010*) implemented on the Galaxy platform (*Afgan et al., 2018*).

## Non-coding RNA analysis

The 'gene_biotype' annotation associated with each Ensembl gene ID was obtained through bio-maRt. Biotypes include lincRNA, antisense RNA, transcribed unitary pseudogenes, transcribed unprocessed pseudogenes, processed transcripts, and protein coding genes. We focused on four gene types: protein coding genes, lincRNA and antisense transcripts given that the other categories have fewer than 50 instances in our data set. We determined the proportion of each gene type in our four response categories as previously described.

## Gene ontology analysis

The gene sets belonging to each of the four response categories were investigated for common pathway enrichment using the KEGG database (*Kanehisa et al., 2017*), within the DAVID genomic annotation tool (*Huang et al., 2009b*; *Huang et al., 2009a*). Enrichment was calculated relative to the set of all 11,974 expressed genes. Multiple testing was performed by the Benjamini-Hochberg method. Pathways enriched at 10% FDR are considered to be significant.

## Cardiovascular gene analysis

A set of 5,010 genes implicated in cardiovascular development or disease (BHF-UCL gene association file) was obtained from the Cardiovascular Gene Ontology Annotation Initiative (https://www.ebi.ac.uk/GOA/CVI). The 2,756 genes that are expressed in our study were overlapped with our four response categories. We determined the proportion of cardiovascular genes in our four response categories as previously described.

## eQTL analysis

For the overlap of our response categories with eGenes from healthy individuals, the list of eGenes in 14 GTEx tissues was downloaded from v7 in the GTEx portal (www.gtexportal.org). eGenes were selected at 5% FDR in each tissue. eGenes from iPSC-derived cardiomyocytes were obtained from *Banovich et al. (2018)* (10% FDR). The eQTL effect sizes of eGenes are defined and reported as allelic fold change by the GTEx consortium.

For the overlap of our response categories with eQTLs identified pre- and post-ischemia, an RNA-seq study of 114 patients undergoing aortic valve replacement surgery was used (*Stone et al., 2019*). In this study, samples of heart left ventricle were obtained for each individual pre- and post-cardioplegic arrest/ischemia. We obtained lists corresponding to all eGenes (5% FDR) identified in males and females combined within the pre-ischemic state (496 expressed in our data), all eGenes in males and females combined in the post-ischemic state (416 expressed), and differentially expressed genes (5% FDR) between pre- and post-ischemia (6,571 in 46 females, and 6,572 in 68 males). Genes that are differentially expressed between pre- and post-ischemia in both males and females (5,115) were used to calculate enrichment in hypoxia response categories. We determined the proportion of eGenes in our four response categories as previously described.

## Genetic tolerance analysis

Existing gene lists, related to genetic tolerance, were overlapped with our hypoxia response genes. The following gene lists were obtained from the Macarthur lab (https://github.com/macarthur-lab/gene_lists): genes nearest to GWAS peaks (*MacArthur et al., 2017*), loss-of-function intolerant genes, and pLI scores (*Lek et al., 2016*). We determined the proportion of gene in each gene set in our four response categories as previously described.

## Data access

All RNA-seq data have been deposited in the Gene Expression Omnibus (www.ncbi.nlm.nih.gov/geo/) under accession number GSE117192.

## Acknowledgements

We thank all members of the Gilad lab for helpful discussions, Kristen Patterson for experimental assistance, and the Genomics Core Facility at the University of Chicago for sequencing the RNA-seq libraries. We thank The Genotype-Tissue Expression (GTEx) Project, supported by the Common Fund of the Office of the Director of the National Institutes of Health, and by NCI, NHGRI, NHLBI, NIDA, NIMH, and NINDS, for providing data. The data used for the analyses described in this manuscript were obtained from the GTEx portal v7 on May 24th 2018.

## Additional information

### Funding

| Funder | Grant reference number | Author |
|---|---|---|
| National Heart, Lung, and Blood Institute | HL092206 | Yoav Gilad |
| EMBO | Long-Term Fellowship ALTF 751-2014 | Michelle C Ward |

The funders had no role in study design, data collection and interpretation, or the decision to submit the work for publication.

### Author contributions

Michelle C Ward, Conceptualization, Formal analysis, Investigation, Visualization, Methodology, Writing—original draft, Project administration; Yoav Gilad, Conceptualization, Supervision, Funding acquisition, Writing—review and editing

### Author ORCIDs

Michelle C Ward    http://orcid.org/0000-0003-1485-320X
Yoav Gilad    https://orcid.org/0000-0001-8284-8926

### Decision letter and Author response

Decision letter https://doi.org/10.7554/eLife.42374.045
Author response https://doi.org/10.7554/eLife.42374.046

## Additional files

### Supplementary files

• Supplementary file 1. Document containing Tables S1-3.
DOI: https://doi.org/10.7554/eLife.42374.031

• Supplementary file 2. Table with species-by-condition interaction genes.
DOI: https://doi.org/10.7554/eLife.42374.032

• Supplementary file 3. Table with all genes assigned to the four gene expression response categories.

DOI: https://doi.org/10.7554/eLife.42374.033

• Transparent reporting form
DOI: https://doi.org/10.7554/eLife.42374.034

### Data availability

Sequencing data have been deposited in GEO under accession code GSE117192.

The following dataset was generated:

| Author(s) | Year | Dataset title | Dataset URL | Database and Identifier |
|---|---|---|---|---|
| Michelle C Ward | 2019 | A generally conserved response to hypoxia in iPSC-derived cardiomyocytes from humans and chimpanzees | https://www.ncbi.nlm.nih.gov/geo/query/acc.cgi?&acc=GSE117192 | NCBI Gene Expression Omnibus, GSE117192 |

The following previously published datasets were used:

| Author(s) | Year | Dataset title | Dataset URL | Database and Identifier |
|---|---|---|---|---|
| GTEx consortium | 2018 | GTEx Analysis v7 eQTL data | https://www.gtexportal.org/home/datasets | Genotype-Tissue Expression Project, GTEx v7 |
| Cardiovascular Gene Ontology Annotation Initiative | 2015 | BHF-UCL gene association file | ftp://ftp.ebi.ac.uk/pub/databases/GO/goa/old/bhf-ucl/ | Gene Ontology Annotation, gene_association.goa_bhf-ucl |
| MacArthur | 2017 | NHGRI-EBI GWAS catalog: Genes nearest to GWAS peaks | https://github.com/macarthur-lab/gene_lists/blob/master/lists/gwas-catalog.tsv | Genes nearest to GWAS peaks, gwascatalog.tsv |

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
