## [Decision Letter]

Thank you for submitting your article "Inter-species differences in response to hypoxia in iPSC-derived cardiomyocytes from humans and chimpanzees" for consideration by *eLife*. Your article has been reviewed by three peer reviewers, one of whom is a member of our Board of Reviewing Editors, and the evaluation has been overseen by Aviv Regev as the Senior Editor. The following individual involved in the review of your submission has agreed to reveal his identity: Charles G Danko (Reviewer #2).

The reviewers have discussed the reviews with one another and the Reviewing Editor has drafted this decision to help you prepare a revised submission.

Summary:

In this paper, the authors perform a comparative study of transcriptional and physiological responses of iPSC derived cardiomyocytes to hypoxia (oxygen deprivation) in human and chimpanzees. Because oxygen deprivation is a major feature of myocardial ischemia, the authors focus on dynamic oxygen response. By performing a comparative analysis of how humans and chimpanzees respond to hypoxia one can gain insight into the susceptibility of humans to different cardiovascular diseases. They measure physiological as well as transcriptional responses in both species under normal oxygen, oxygen deprivation, reoxygenation and perform a careful comparative analysis of the gene expression differences and similarities between the two species.

This is well designed and executed study and manuscript is very well written and clear. The overall conclusions of the paper is that response to hypoxia is generally conserved across species and might be less tolerant for genetic variation. Genes that respond to hypoxia are less tolerant of genetic changes as they are depleted for eQTLs associated with heart and enriched for genes intolerant for loss of function. The study also finds several key genes that are species-specific (e.g., 147 genes, or 199 in human and 433 in chimpanzee depending on the analysis), including RASD1, which is also upregulated in patients post-ischemia and falls in a locus associated with ischemic stroke.

Overall, this is a very interesting study which has important implications for the stress response field. However, there are some aspects of the manuscript and analysis performed that need improvements to (a) make the connection to the eQTL analysis earlier on in the paper and (b) to increase the depth and scope of this study to gain more insight into the species-specific differences.

Essential revisions:

1) The Introduction focuses on applying a comparative approach to study species-specific heart disease vulnerabilities, which is an important question that can be tackled by the authors' innovative strategy. The Introduction should be expanded to further explain whether the species differences in heart disease susceptibility are thought to be genetic or environmental, and which species is thought to have the derived disease risk change. Fleshing these points out further would help to clarify how gene expression and genome sequence differences could inform species disease predisposition.

2) Despite the Introduction focused on comparative vulnerabilities, the Results and Discussion provide equal or greater emphasis on the point that eQTL genes are depleted from the stress response. This finding seems to play into a larger debate (introduced in the Discussion) about whether eQTLs represent disease associated variation or are more likely to reflect neutral genetic variants, but the significance of this debate is not introduced in advance. If the properties of genes with conserved responses to oxygen deprivation is to be a major focus of the paper, then the Introduction should build out more of the significance of this problem and the logic of why the comparative strategy can help address this question. Could the same question have been asked without comparative data, or is the filter for conserved genes necessary?

3) Although the responses to hypoxia are conserved, additional work is needed to increase the scope of the paper and in particular examine the species-specific differences. In particular, the authors identify a set of species specific genes which already include good candidates (e.g., RASD1 in human, TGFbeta genes in chimp) that may relate to disease vulnerabilities, but the authors do not pursue this analysis further. Can the authors validate by Western blot that RASD1 (or other candidates) are indeed up-regulated only in the response of human cells (in a subset of individuals)? Some form of validation would be important if not a functional analysis of top candidates.

4) Alternatively, to further explore the species specific differences the authors could leverage published or newly collected datasets measuring the binding profile of the key transcription factors on the regulatory regions of the responsive genes and see if there are differences at the level of regulation. The transcription factors driving hypoxia (e.g., HIF1 and FOXO1) have been previously characterized at the level of either ChIP-seq or binding motifs. The authors could use published ChIP-seq datasets and ask whether peaks near conserved response genes show greater sequence conservation than peaks near human-specific response genes. This would indicate a potential evolutionary mechanism for divergent response patterns. Similarly, for high priority candidates like RASD1, the authors could directly analyze whether humans have gained HIF1 or FOXO1 binding motifs.

5) Previous work from the same lab (Pavlovic et al., 2018) showed strong correspondence between in vitro derived cardiomyocotyes and human heart tissue. However, one of the major differences reported between iPSC-CMs and heart tissue related to oxidation-reduction processes. How representative is the in vitro response likely to be given these differences? Clearly, the authors still observe major hypoxia responses (and formulated low glucose media conditions to enable this), but can the authors clarify any limitations based on their past observation?

6) The analysis seems to be restricted to one-to-one orthologs. Could more interesting and interpretable differences be identified if one was to take the gene duplications into account? The depth of the study could also be increased by examining annotated lincRNAs.

---

## [Author Response]

Essential revisions:1) The Introduction focuses on applying a comparative approach to study species-specific heart disease vulnerabilities, which is an important question that can be tackled by the authors' innovative strategy. The Introduction should be expanded to further explain whether the species differences in heart disease susceptibility are thought to be genetic or environmental, and which species is thought to have the derived disease risk change. Fleshing these points out further would help to clarify how gene expression and genome sequence differences could inform species disease predisposition.

We have added further evolutionary context in the second paragraph of the Introduction. Briefly, we now discuss anatomical level differences between the species, such as the interstitial fibrosis aetiology, which is prevalent in chimpanzees and is also the predominantly diagnosed form of cardiovascular disease (CVD) in captive bonobos, gorillas and orangutans (Lowenstine et al., 2015).

It should be noted that there is limited data on the disease in chimpanzees compared to humans, as might be expected, and to our knowledge there is no genetic data about differences in susceptibility to CVD between the two species.

2) Despite the Introduction focused on comparative vulnerabilities, the Results and Discussion provide equal or greater emphasis on the point that eQTL genes are depleted from the stress response. This finding seems to play into a larger debate (introduced in the Discussion) about whether eQTLs represent disease associated variation or are more likely to reflect neutral genetic variants, but the significance of this debate is not introduced in advance. If the properties of genes with conserved responses to oxygen deprivation is to be a major focus of the paper, then the Introduction should build out more of the significance of this problem and the logic of why the comparative strategy can help address this question. Could the same question have been asked without comparative data, or is the filter for conserved genes necessary?

We have revised the first paragraph of the Introduction to provide rationale behind our chosen approach, and to outline an alternative within-species approach to understanding complex traits and disease.

Indeed, we agree that this is one of the most interesting findings in the paper; however this was not one of the original goals of our study. Once we identified clusters of genes responding in a conserved versus species-specific manner, we realized that we had the opportunity to investigate different properties associated with the response. One property of interest was to consider genes whose expression level is variable across individuals i.e. eQTLs. Our conclusions with respect to eQTLs would have been the same even if we only had the human data. This, however, may not be true for responses to other treatments because in our study most genes respond to hypoxia in both species (1,920 conserved response genes and 630 species-specific response genes). It is not clear if this is a general property that would extend to other stress responses.

3) Although the responses to hypoxia are conserved, additional work is needed to increase the scope of the paper and in particular examine the species-specific differences. In particular, the authors identify a set of species specific genes which already include good candidates (e.g., RASD1 in human, TGFbeta genes in chimp) that may relate to disease vulnerabilities, but the authors do not pursue this analysis further. Can the authors validate by Western blot that RASD1 (or other candidates) are indeed up-regulated only in the response of human cells (in a subset of individuals)? Some form of validation would be important if not a functional analysis of top candidates.

It is important for us to note that testing for different levels of responses should not be considered ‘validation’ of the observation at the RNA level. Protein levels, for example, can and often will show patterns that are not consistent with the RNA level. We prefer the terminology of testing whether the RNA response we observed extends to other levels as well.

Along these lines, we have now performed several additional experiments in order to increase the scope of the paper. We first considered cellular phenotypes that are likely to be impacted by hypoxic stress, and may show a difference between species based on inter-species differences in the disease phenotype. We determined sensitivity to oxygen deprivation by measuring the level of cytotoxicity during the course of the experiment. A hallmark of cellular toxicity is the permeabilisation of the outer cellular membrane resulting in the release of intracellular components into the surrounding milieu. The activity of the lactate dehydrogenase (LDH) enzyme, which interconverts pyruvate and lactate, can be measured in the cell culture media as a proxy of this process. We observed a marginal yet significant increase in LDH activity following hypoxia in humans, and a significant increase following short-term re-oxygenation in both species. Despite these apparent differences in response to hypoxia, there is no significant difference in LDH activity between species within a condition. We have included these results as Figure 8A, and described them in the first paragraph of the subsection “Cellular responses to hypoxia”.

We then asked whether oxygen deprivation experienced by cardiomyocytes could have downstream consequences on other cell types in the heart such as cardiac fibroblasts by secretion of cytokine signaling molecules. The TGFβ-1 cytokine is known to mediate the development of fibrosis following stress in the heart (Liu et al., 2017). We measured secreted TGFβ-1 protein levels by ELISA for four individuals in each species. We found that TGFβ-1 release was significantly increased following re-oxygenation after hypoxic stress in both species; however there is no difference between species under any condition. We have included these results as Figure 8B, and added a description of these results in the last paragraph of the subsection “Cellular responses to hypoxia”.

As the additional assays we performed to measure cellular phenotypes in response to hypoxic stress again show conservation across species, we have reworded the title to more accurately reflect the fact that while we have identified over 100 species-specific response genes through a stringent interaction analysis, the response is largely conserved across species.

In order to further explore species-specific differences, we attempted to measure protein expression of four species-specific response genes identified through our stringent species-by-condition interaction approach. Given the small number of cells remaining from the experiment, we were unable to detect protein expression for three antibodies we tested: RASD1, JUNB and BCKDK using 8 or 12 μg protein, and two antibody conditions. We were able to detect SNIP1 protein expression in two individuals of each species, but we question whether we detected the correct isoform.

Unfortunately, our study was not designed to measure protein expression, and all optimization to determine the media conditions and timepoints in the study was assessed using mRNA expression. It is not practical at this time repeat the entire experiment in order to collect more protein data. We provide the western blot analysis in our response (Author response image 1), but we decided not to include this in the manuscript, as we don’t believe that these results add meaningfully to the narrative.

Finally, we attempted to assess the functional consequences of the RASD1 GTPase, and other G-protein related species-specific response genes by measuring the level of cAMP secretion, a downstream effector of G-protein signaling, via an ELISA assay. In our hands we did not find the assay to be sensitive enough to consistently detect cAMP levels across a test set of samples, and therefore did not perform this experiment on our full set of samples.

4) Alternatively, to further explore the species specific differences the authors could leverage published or newly collected datasets measuring the binding profile of the key transcription factors on the regulatory regions of the responsive genes and see if there are differences at the level of regulation. The transcription factors driving hypoxia (e.g., HIF1 and FOXO1) have been previously characterized at the level of either ChIP-seq or binding motifs. The authors could use published ChIP-seq datasets and ask whether peaks near conserved response genes show greater sequence conservation than peaks near human-specific response genes. This would indicate a potential evolutionary mechanism for divergent response patterns. Similarly, for high priority candidates like RASD1, the authors could directly analyze whether humans have gained HIF1 or FOXO1 binding motifs.

This is a good idea. Thank you. We obtained published human ChIP-seq data sets for HIF1α (356 sites), HIF2α (301 sites) (Schodel et al., 2011), and FOXO3 (934 sites) (Eijkelenboom et al., 2013). HIF1/2α binding was measured in a breast cancer cell line following HIF stabilization with dimethyloxaloylglycine. FOXO3 binding was measured in a colon cancer cell line with a Tamoxifen-inducible FOXO3A3-ER fusion protein. We performed combined analysis of these data with our gene expression data, and we believe that the results do add a bit of insight into potential mechanisms that underlie the observed response. These results are now reported in the second and third paragraphs of the subsection “Properties of hypoxic response genes” and in Figures 6A-B.

5) Previous work from the same lab (Pavlovic et al., 2018) showed strong correspondence between in vitro derived cardiomyocotyes and human heart tissue. However, one of the major differences reported between iPSC-CMs and heart tissue related to oxidation-reduction processes. How representative is the in vitro response likely to be given these differences? Clearly, the authors still observe major hypoxia responses (and formulated low glucose media conditions to enable this), but can the authors clarify any limitations based on their past observation?

Yes, we should have obviously done that. Thank you for pointing this out.

We overlapped our four response gene categories with the set of genes that Pavlovic *et al.* found to be differentially expressed between heart tissue and iPSC-derived cardiomyocytes at differentiation Day 27. Out of 2,459 genes that respond to hypoxia in one or both species, only 371 are differentially expressed between hearts and iPSC-derived cardiomyocytes (16%). 34 of the 147 species-by-condition interaction genes (23%) are differentially expressed between hearts and iPSC-derived cardiomyocytes. This suggests that the majority of response genes are not differentially expressed between hearts and iPSC-derived cardiomyocytes, and that iPSC-derived cardiomyocytes are a useful model for studying in vivo processes. We have included this description in the first paragraph of the Discussion subsection “Potential limitations of our model”.

We also now also note in the Discussion (second paragraph), that the heart is a complex tissue consisting of multiple cell types and these other cell types, with alternative metabolic profiles, will contribute to the signal in heart tissue, which will be absent from the in vitro derived cardiomyocytes.

6) The analysis seems to be restricted to one-to-one orthologs. Could more interesting and interpretable differences be identified if one was to take the gene duplications into account? The depth of the study could also be increased by examining annotated lincRNAs.

We have now added an analysis of non-coding transcripts. We found that antisense transcripts are enriched in conserved response genes compared to non-response genes; however, species-specific response genes are no more likely to be antisense transcripts than conserved response genes. When we focused on the more stringent set of species-by-condition interaction genes, we found an enrichment of lincRNAs. These lincRNAs are *APTR, NEAT1, RNF139-AS1* and *LINC02615*. We have included these results as Figure 6C, and they are described in the fourth paragraph of the subsection “Properties of hypoxic response genes”.

With respect to orthology, we chose to focus on one-to-one orthologs in order to have a comparable set of genes from which to identify gene expression differences. It is not entirely clear to us how to analyze comparative gene expression data if it can be mapped to only a single species. We have specified the rationale behind our chosen approach in the first paragraph of the subsection “RNA-seq data processing”.